# Bacterial cell-size changes resulting from altering the relative expression of Min proteins

Harsh Vashistha[1,3,4], Joanna Jammal-Touma[1,4], Kulveer Singh[2], Yitzhak Rabin[2] & Hanna Salman [1] ✉

The timing of cell division, and thus cell size in bacteria, is determined in part by the accumulation dynamics of the protein FtsZ, which forms the septal ring. FtsZ localization depends on membrane-associated Min proteins, which inhibit FtsZ binding to the cell pole membrane. Changes in the relative concentrations of Min proteins can disrupt FtsZ binding to the membrane, which in turn can delay cell division until a certain cell size is reached, in which the dynamics of Min proteins frees the cell membrane long enough to allow FtsZ ring formation. Here, we study the effect of Min proteins relative expression on the dynamics of FtsZ ring formation and cell size in individual *Escherichia coli* bacteria. Upon inducing overexpression of *minE*, cell size increases gradually to a new steady-state value. Concurrently, the time required to initiate FtsZ ring formation grows as the size approaches the new steady-state, at which point the ring formation initiates as early as before induction. These results highlight the contribution of Min proteins to cell size control, which may be partially responsible for the size fluctuations observed in bacterial populations, and may clarify how the size difference acquired during asymmetric cell division is offset.

Living cells exhibit temporal fluctuations and cell-to-cell variations in all measurable cellular properties, such as cell size and protein content[1–13]. The variations, though, are confined to a restricted range specific to the property being quantified and the environment in which the cells are growing[7], which indicates that cellular properties are subject to strict regulation mechanisms. Uncovering what determines the observed fluctuations in cellular physical and physiological characteristics, such as size and growth rate, is of paramount importance as they affect the cells' ability to carry out their functions. In the case of the bacterium *Escherichia coli (E. coli)*, for example, which is the focus of this study, it has been shown that the fitness of a bacterial community correlates with the population's average cell size in a fluctuating environment[14]. Despite its importance, however, and after decades of extensive research, the molecular mechanisms that control cell size remain under debate.

Significant advancement in understanding cell size control in bacteria has been recently achieved using high-quality single-cell measurements of growth dynamics[6,13,15]. Using simple mapping of the cellular growth and division dynamics, several phenomenological models were developed to explain cell size control in bacteria[3,13,15–25]. These models are very successful in explaining experimental observations at the population level. For example, the widely accepted model of size homeostasis, the adder model, proposes that cells add a constant volume during their cell cycles irrespective of their birth size, which allows them to correct for size fluctuations over time. While the mechanisms underlying size control in bacteria are still under investigation, recent studies have revealed key molecular factors and cellular processes that contribute to size homeostasis[26–35]. The prevailing picture that emerges from these studies is that of a complex

[1]Department of Physics and Astronomy, University of Pittsburgh, Pittsburgh, PA, USA. [2]Department of Physics and Institute for Nanotechnology and Advanced Materials, Bar-Ilan University, Ramat-Gan, Israel. [3]Present address: Department of Molecular, Cellular and Developmental Biology, Yale University, New Haven, CT, USA. [4]These author contributed equally: Harsh Vashistha, Joanna Jammal-Touma. ✉e-mail: hsalman@pitt.edu

coordination between two processes that need to be completed between consecutive division events, which together determine the time of division and the size added during the cell cycle. The first is DNA replication, while the second is arguably independent of replication[30,32–34]. In slow growth conditions, the replication process would be a limiting factor for the division events[31,35], while in fast growth conditions where multiple replication forks proceed concurrently, DNA replication becomes less important in determining the cell division timing[19,31]. In a recent study, it was suggested that the proteins required for septum formation need to accumulate to a threshold amount to be able to initiate the constriction[19]. This also provides a mechanistic foundation for the adder phenomenon. The study argued compellingly that the added size can be simply controlled by requiring that a specific protein in the cell accumulates to a threshold amount before the cell can divide. Since the protein amount and not its concentration is the determining factor for division, such a requirement removes any dependency on cell size, which affects protein concentration. This protein was shown to be the FtsZ, which forms the septal ring and drives cell division. As such, the amount of FtsZ in the cell needs to reach a certain threshold in order to complete the septal ring and initiate self-contraction to form new cellular poles and divide the cell into two[36,37]. Indeed, the study found that FtsZ accumulation in the cell exhibits a strong correlation to the added volume and thus supports a mechanism in which the added size is determined by the need to accumulate a threshold quantity of FtsZ[19]. This is further supported by other studies, which demonstrated that reducing the amount of FtsZ[29], increasing its degradation rate[38], or increasing the cell width[29,32] (which would increase the amount of FtsZ needed to complete the septum formation) delayed cell division.

However, FtsZ does not operate independently in the cell. It has been demonstrated in many studies that the placement of FtsZ in the cell is determined in part by oscillatory dynamics of membrane-associated proteins, collectively termed the Min system[39–47] (see Haeusser & Margolin, and Rowlett & Margolin[48,49] and references therein for a comprehensive picture of the regulation and dynamics of the septal ring formation in bacteria). One of these proteins, MinD, can be found in ATP- and ADP-associated forms (MinD-ATP and MinD-ADP, respectively). MinD-ATP binds to the cell membrane and recruits MinC, which in turn prevents the binding of FtsZ to the membrane. A third Min protein, namely MinE, chases the MinCD complex and hydrolyzes MinD-ATP, which breaks this complex and causes it to dissociate from the membrane. Free MinD-ADP and MinC can then diffuse in the cytoplasm, where MinD-ADP can be converted back to MinD-ATP and bind to the membrane at different locations. This interplay between the proteins of the Min system has been shown to create surface waves in vitro[50], with a specific wavelength that depends on the relative concentrations of the participating components. In living wild-type cells, this behavior is translated into pole-to-pole oscillations of these proteins that occur on a timescale of ~40 s/oscillation[42–44,51,52]. The MinCD wave sweeping of the cell prevents FtsZ-membrane binding at its path. When averaged over time, the concentration of MinC in the cell forms a nonlinear gradient with maximal concentration at the poles and minimal concentration away from the poles. This allows FtsZ to bind and form the septal ring somewhere along the cell membrane, away from the poles and close to the mid-cell, under natural growth conditions.

These previous findings, both in vitro and in vivo, indicate that the dynamics of the Min proteins have an intrinsic pattern that depends on factors like protein concentration, reaction rates, and geometry[53–56]. In addition, the Min pole-to-pole oscillations have been shown to affect the timing of cell division in E. coli[57]. These observations suggest that the balance between the different Min proteins can modulate the timing of the FtsZ ring placement to coincide with the cell reaching a length that allows for a dynamical pattern of the Min proteins that will not disrupt the FtsZ ring formation. This, in turn, determines the initial

cell size, to which the cell will add a constant volume as dictated by the FtsZ accumulation to a threshold amount and result in additional variation in cell size. Support for this scheme was provided by recent studies, which showed that a sister cell that receives a smaller fraction of the mother at cell division adds more volume during the first cell cycle following the division compared to its larger sibling[3,58]. Assuming that both sister cells receive similar ratios of MinE/MinD, the smaller sister then needs to grow slightly more before a Min dynamical pattern is created that would leave the membrane free long enough to allow for continuous FtsZ accumulation and ring formation.

In this work, we track cell size changes in response to varying the relative expression of the Min proteins under relatively fast growth conditions, where replication is not a limiting factor to division, in order to further probe and verify the role of the Min dynamics in cell-size control. We find that changing the relative expression of Min proteins affects the average cell size of the bacterial population. Our analyses at the single-cell level reveal that the altered ratio of Min proteins, specifically increasing MinE/MinD, delays the FtsZ ring formation, which in turn allows the cell to grow for a longer time and reach a larger size. We hypothesize that the delay in the FtsZ ring formation is a result of the disruption of FtsZ accumulation at the membrane by the Min proteins oscillations, whose dynamical pattern depends on the cell length. Once the cell reaches a length that facilitates a regular dynamical pattern of the Min proteins, which permits a continuous accumulation of FtsZ at the membrane, a stable septal ring that does not disintegrate forms and grows until division. A comparison of our measurements with the predictions of a simple theoretical model of the Min dynamics provides support to our hypothesis. These results indicate that the Min proteins can affect cell size in E. coli by regulating the initiation time of the FtsZ ring formation.

## Results

### Increased expression of MinE increases cell size
We began by testing how altering the MinE/MinD expression ratio affects the average cell size compared to natural conditions. The ratio MinE/MinD was altered by overexpressing either MinE or MinD in the cells. This was achieved by transforming the cells with plasmids containing arabinose-inducible promoter controlling the expression of one of the two genes, minE–mEos or mEos–minD, which produce MinE or MinD proteins fused with the fluorescent protein mEos. Using these constructs, we were able to regulate the expression level of minE or minD beyond the wild-type expression level by inducing the promoter with different arabinose concentrations (Fig. S1, see "Material and methods" for experimental details).

Images of the cells grown at several different expression levels were acquired, and the population's average cell size for cells over-expressing minE (Fig. 1A) or minD (Fig. 1B) was calculated. Our results show that overexpressing minD or minE resulted in an increase in the population's average cell size. Cells overexpressing minD showed a continued increase in average cell size while those overexpressing minE saturated to a new size at high inducer concentration. In the case of minD overexpression, in addition to becoming uncontrollably long, MinD appeared to occupy most of the cellular membrane and attached to the cell membrane in random patches (Fig. 1C). This provides an insight into why cells were growing uncontrollably long at high inducer concentration. MinD is known to recruit MinC and prevent the FtsZ binding to the membrane and thus inhibit cell division. It has also been suggested that proper recruitment of MinD and its organization onto the membrane requires MinE[41]. Hence, the presence of MinD in excess amount, while MinE is maintained at its natural expression level, would lead to disordered accumulation of MinD on the cell membrane and hinder the Min oscillations and, therefore, cell division. Due to the extreme effect of overexpressing minD, we limit our investigation to what follows minE overexpression.

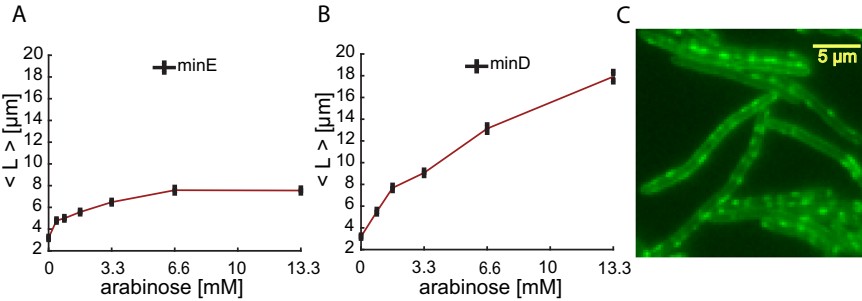

**Fig. 1 | The effect of overexpressing *minE* or *minD* on the average cell size.** The average cell size in a population (obtained from an image of 500 individual cells) in which *minE* (**A**) or *minD* (**B**) overexpression was induced from P$_{araBAD}$ to various levels using different inducer (arabinose) concentrations as indicated on the *x*-axes.

**C** Fluorescent image of mEos−MinD in filamented cells, in which *minD* was over-expressed, showing the random nature of MinD binding to the membrane. Source data are provided as a Source Data file.

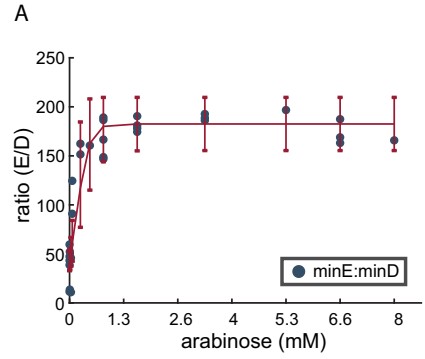

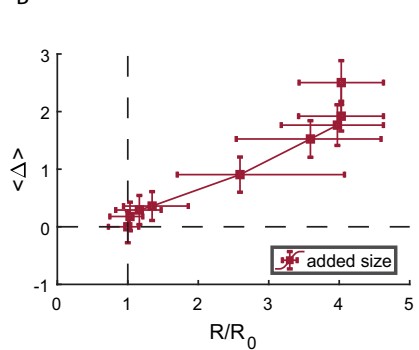

**Fig. 2 | The effect of MinE/MinD ratio on the average cell size in a population.**
**A** The ratio of MinE/MinD was measured using quantitative real-time PCR (see "Materials and methods") as a function of the inducer (arabinose) concentration. Each point in the graph represents the average of three measurements carried out simultaneously, and the error bars represent the fit error. **B** The average increase in the size of cells under the different concentrations of inducer (size distributions are presented in Fig. S2), measured from images of the cells (500 cells/condition), is plotted as a function of the MinE/MinD ratio (*R*) relative to the ratio in wild-type bacteria prior to the overexpression of *minE* ($R_0$). The ratios were calculated from the fit in (**A**): $R = (2.878 \mp 11.296) + \frac{179.63 \mp 15.82}{1 + \text{Exp}(-(C - (0.182 \mp 0.07))/(0.158 \mp 0.077))}$, where *C* is the Arabinose concentration. The values of the inflection point and rate of increase in

the fit are exactly the same as those obtained from the independent measurements of *gfp* expression from the same promoter P$_{araBAD}$ presented in Fig. S3. Note that the increase in the *minE/minD* expression ratio is capped by what the induced promoter allows (**A**). Therefore, we do not know whether the average cell size will keep increasing or will saturate for larger $R/R_0$ values (**B**). It is possible that much larger ratios will block cell division completely and cause uncontrolled cell filamentation, which would suggest that the graph in (**B**) will exhibit divergence for larger $R/R_0$ values. Error bars in $R/R_0$ are calculated from fit error, and the error bars in the size increase represent the standard error determined from the size variation in the measurements presented in Fig. S2. Source data are provided as a Source Data file.

To quantify the level of added expression of *minE*, we measured the expression ratio *minE/minD* at the mRNA level using real-time quantitative PCR (RT-qPCR) as detailed in "Materials and methods." The use of RT-qPCR for measuring expression ratio circumvents the need for labeling both proteins in the same cell and measuring their fluorescence simultaneously. This also allows us to preserve the native *minE* and *minD* genes in the cell and reduce external interference to a minimum. The mRNA was extracted from cultures and induced with different arabinose concentrations to express *minE−mEos* at different levels. Our results show that the expression ratio *minE/minD* increases with the increase in inducer concentration (Fig. 2A). Note that while the ratio of the mRNA might not be identical to the proteins' ratio, the two ratios are linearly proportional to each other. Therefore, we can accurately determine how this ratio changes relative to a fixed reference condition, which we take here to be the uninduced condition. Plotting the average population size as a function of the change in the expression ratio *minE/minD* relative to the uninduced ratio (Fig. 2B) confirms its effect on the population's average cell size (see Fig. S2 for the size distributions at the different induction levels).

These results clearly demonstrate that the expression ratio *minE/minD* affects the cell size. We argue that this effect is mediated by the need to have an appropriate cell size, in which the Min protein's

dynamical pattern allows uninterrupted FtsZ membrane binding early in the cell cycle. This size will change with the relative concentrations of the Min proteins. Thus, the cell size at birth is one in which the existing concentrations of Min proteins in the cell can produce stable patterns that permit stable FtsZ ring formation early in the cell cycle. Altering the concentration of one of these proteins would then change the dynamical pattern and frequency of the MinCD wave sweeping the cell length[50] (see Supplementary Movies 2 and 3) and, therefore, prevent uninterrupted FtsZ binding to the membrane at that cell size and delay cell division until the cell reaches a new length, in which the Min proteins dynamical pattern with the new concentrations allows the formation of a stable FtsZ ring. To verify this hypothesis, one needs to probe the dynamics of cell size change and FtsZ membrane-binding simultaneously, following the induction of *minE* overexpression, which we examine in the subsequent section.

### Effect of MinE overexpression on the binding time of FtsZ
Here we address the question: how does modifying the Min proteins concentrations balance lead to cell size alteration? We recall that FtsZ was shown to bind to the membrane within the first quarter of the cell cycle[37,59,60]. This suggests that the septum ring localization and FtsZ binding to the membrane should be enabled by the Min dynamical

pattern early in the cell cycle. Thus, when the concentration balance of the Min proteins is modified, FtsZ binding is expected to be disrupted until the cell reaches a size in which the Min proteins with the new concentrations can generate a dynamical pattern that allows FtsZ to stably accumulate at the membrane. Once FtsZ accumulation at the membrane is uninterrupted, the ring becomes stable and grows continuously, and the cell will grow until the FtsZ ring can initiate constriction and cell division. During this last stage of the process, while the FtsZ ring is being completed, the exponential size increase is expected to be constant as the time for ring completion should remain fixed. The extra growth prior to the stable Z-ring formation will therefore lead to an increase in cell size following the alteration of the Min proteins balance, and as a result, larger daughter cells will be generated upon cell division. In the new larger daughter cells, however, the Min dynamical pattern that allows FtsZ membrane binding and stable Z-ring formation should be generated earlier than in the previous cell cycle, and the FtsZ ring formation will be initiated earlier as well. This implies that once the Min proteins reach their final stable concentrations following the induced increase in *minE* expression, the FtsZ ring formation should initiate as early as it did before the increased expression of *minE*. In other words, while the new daughter cell size increases gradually and monotonically following the increase in *minE* expression until it reaches the new steady-state size, the delay in FtsZ ring formation should be transient, and once the new steady-state is achieved, it will occur as early as for the previous steady-state size.

To verify this hypothesis and investigate the mechanism through which *minE* expression amplification enlarges the cell size, we probed the FtsZ ring formation dynamics simultaneously with cell size following *minE* overexpression induction. This was achieved using the bacterial strain KC376, which is an MG1655 derivative containing the fluorescent protein, mVenus, integrated within the chromosomal *ftsZ* gene[61]. This integration produces a functional fluorescent FtsZ protein, and this strain has been used previously to visualize the Z ring dynamics and does not impair bacterial cell division[19] (Fig. 3A).

The KC376 strain was also transformed with two additional plasmids expressing *mCherry* constitutively and *minE–mEos* under the control of P*araBAD* (see "Materials and methods"). The additional plasmids served to help better visualize the cells' boundaries in the microfluidic traps during the experiments and to control the overexpression of *minE*, respectively. The new cells were grown in the mother machine, where cell size and the FtsZ ring could be monitored simultaneously in individual cells. After several hours of growth with a natural *minE* expression level, the expression of *minE–mEos* was induced by adding arabinose to the feeding solution (see "Materials and methods" for further details). Our observations show that immediately following the induction of *minE–mEos* expression, cells grow to a longer size than in previous cell cycles before dividing (Fig. 3A, B and Supplementary Movie 1), and their cell cycle duration is extended (Fig. 3B). Additionally, the stabilization of the FtsZ ring at the membrane exhibited significant delay (Fig. 3C, D). These results demonstrate that an increase in the ratio of MinE/MinD disrupts the normal FtsZ ring dynamics in the induced cells.

Further examination of cell size dynamics and FtsZ ring formation (illustrative examples of the measurements are presented in Fig. S4) reveals several important features of cell size increase. Consistent with our earlier prediction, we observe that both the time and size required to form a stable FtsZ ring initially increase following the rise in *minE* expression (Fig. 4A–C). Within ~4 generations, on average, after the induction of *minE* overexpression, the time required for stable ring formation decreases back to its initial value observed prior to induction (Fig. 4B). The size required for achieving stable FtsZ-ring, however, gradually stabilizes at a higher value than observed under uninduced *minE* expression level (Fig. 4C).

Moreover, the time required to reach division once the FtsZ is stably assembled at mid-cell remains constant throughout the

transition from one steady-state cell size to another (Fig. 4D). Correspondingly, the exponential increase in size during this time also remains constant as predicted earlier (Fig. 4E). Note that the induction level of *minE* overexpression and the growth medium do not affect the observed dynamics presented here. Changing the induction level of *minE* overexpression, or the growth medium, changes only the values of the stable cell size before and after the induction. The dynamics of the measured parameters depicted in Fig. 4F, G remain unchanged (Fig. S5). This is to be expected because the induction dynamics of the P*araBAD* promoter are the same for all induction levels (see section of model predictions and Fig. S3). Additionally, the stable cell size following the induction was found to be proportional to the induction level of *minE* (Fig. S6 insets and Fig. S7), as we discuss in the following section.

These observations indicate that the delay in FtsZ binding at mid-cell (Fig. S8) causes the cell to grow for a longer time before dividing, which in turn results in an increase in cell size. Once the cell reaches the new steady-state cell size, the stable accumulation of FtsZ at mid-cell starts early, and therefore, the cell cycle time is no longer extended, and the cell size is maintained from that point on. This result stands in good agreement with previous studies, which suggest that the adder phenomenon is mediated by the FtsZ ring formation dynamics[19]. Our measurements show that the adder correlations are preserved during the induction process of *minE* overexpression and until the cell reaches a new steady state following the induction (Fig. S9).

### Experimental results confirm model predictions

Previous theoretical models have described how the Min proteins can create stable pole-to-pole oscillations in a bacterial cell along its long axis[62,63]. These models have concluded that at the optimal conditions, the dynamics of MinCD binding-unbinding to the cell membrane creates a wave-like pattern[50] that sweeps the cell length with a minimal occupation time of the membrane at the middle of the cell[62,63]. Altering the ratio MinE/MinD will affect the MinCD cell-sweeping pattern and frequency, which can cause the removal of the FtsZ ring before a new sweeping pattern is established that would allow for its stable accumulation and the completion of the septum ring (Fig. 4A middle image).

To further support the experimental results presented in the previous section, we compared the measured cell size and FtsZ ring formation dynamics to an established model prediction. We performed numerical simulations of the Min system dynamics, developed previously by Wehrens et al.[64] based on the model by Huang et al.[63], and reduced here to 1-d (see Supplementary Information and Supplementary Software). The results of simulations carried out in cells of different lengths confirm that the balance between the different ingredients of the Min System and the cell length determines the dynamical pattern of the MinCD wave sweeping the membrane (Supplementary Movies 2 and 3). Typical average profiles of the MinCD concentration along the cell membrane are presented in Fig. S10A–C. It is clear from these profiles that there is an optimal cell length for which the mid-cell has a minimal average concentration of MinCD. Cell lengths smaller or larger than the optimal length exhibit a more uniform average distribution along the cell, resulting in a reduced probability of FtsZ binding to the membrane at mid-cell. This can be also seen by calculating the time during which that location is unoccupied by MinCD (i.e., the number of MinCD molecules there is below a certain threshold) and which is maximized for a specific cell length (Fig. S10D). The longer the time during which this location is free of MinCD, the higher the probability for the FtsZ protein to initiate the septum formation at that location.

To compare model predictions with our experimental results, we carried out simulations for different MinE/MinD ratios and obtained the optimal cell length for each concentration, which would allow for a low enough MinCD occupation time at mid-cell (Fig. 5A). We used linear first-order approximation to describe the optimal cell size as a function of the change in the ratio MinE/MinD.

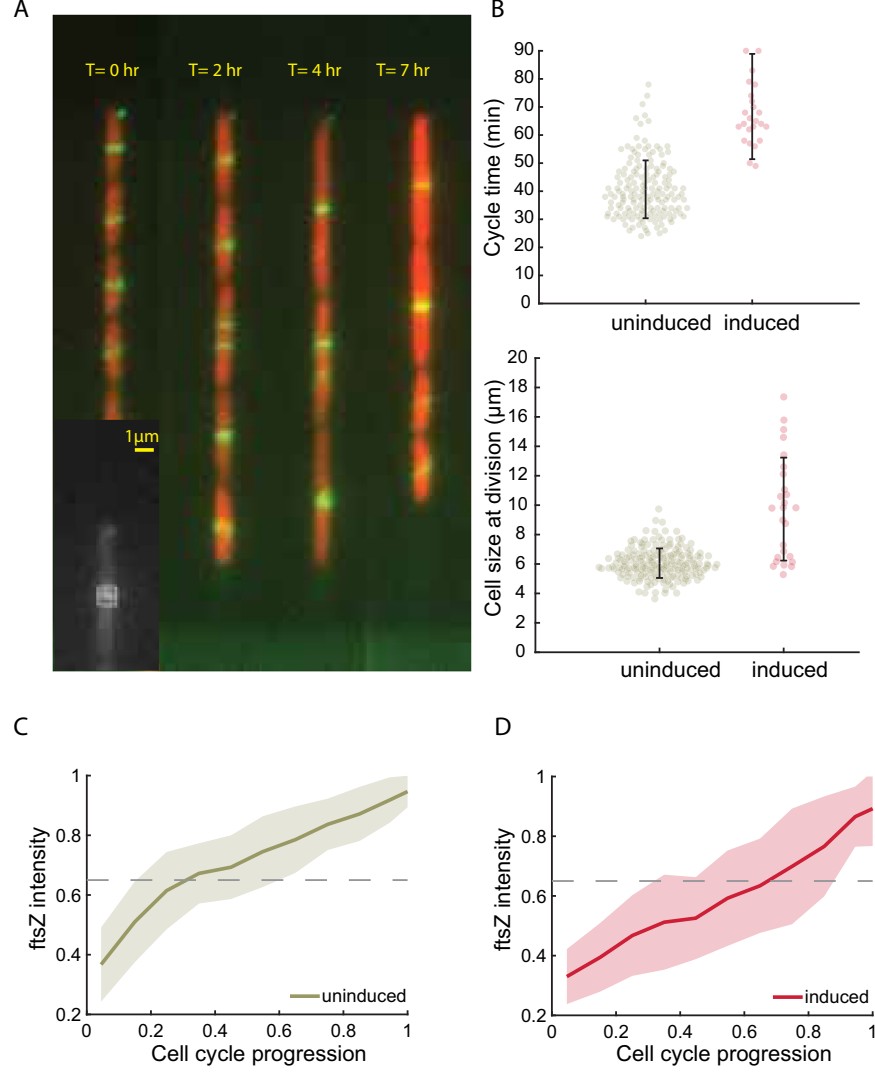

**Fig. 3 | Dynamics of FtsZ ring formation. A** Snapshots of a growth channel taken at different time points in a typical experiment (See Supplementary Movie 1 for the complete experiment), where the FtsZ ring is visible in green. The cell is also expressing mCherry to allow better detection of its boundary. The accumulation kinetics of FtsZ at the septum were evaluated from the fluorescence intensity within a small region around the septum, as highlighted in the inset. **B** The average cell length at division (bottom panel) and cell cycle time (top panel) before inducing the overexpression of *minE* (165 points, uninduced) and during the first cell cycle after inducing minE overexpression, in which cells become elongated relative to preceding cycles (26 points, induced). The error bars represent the standard deviation. **C** FtsZ ring intensity in cells growing under uninduced conditions of *minE* expression reached 65% of the maximum in the first quarter of the cell cycle. **D** After induction with 0.0025% w/v arabinose, the intensity of the FtsZ ring in elongating cells reached 65% of maximum intensity in the last quarter of their cell cycles. The line in **C** and **D** depicts the average of 39 cells, and the shaded area depicts the standard deviation. This shows that membrane-binding of FtsZ is significantly delayed by overexpression of *minE*.

We then examined how this relationship between the MinE/MinD ratio and the optimal cell size affects the kinetics of cell size and FtsZ binding time when increasing the expression of *minE* as we did in the above experiments. When MinE production is induced from $P_{araBAD}$, its increase over time relative to MinD follows the trajectory described by[65] (Fig. 5B):

$$MinE(t) = MinE_{initial} + \frac{(MinE_{final} - MinE_{initial})}{\left(1 + e^{-\frac{t - t_c}{t_s}}\right)} \quad (1)$$

where $MinE_{initial}$ is the initial amount of MinE in the cell prior to induction of overexpression from $P_{araBAD}$, $t$ is the time following induction at $t = 0$, $t_c$ is the time needed to reach ~1/2 of the maximum added MinE ($1/2(MinE_{final} - MinE_{initial})$), and $t_s$ is the time after $t_c$ required to reach ~2/3 of the maximum added MinE. The kinetics of

expression from $P_{araBAD}$ was also confirmed by measuring the expression of *gfp* under its control (Fig. S3).

Using the two curves in Fig. 5A, B, we can calculate the cell length that would facilitate the MinCDE dynamical pattern with minimal MinCD occupancy of the mid-cell at every generation $g$ ($L_{stable}(g)$) following the induction of *minE* overexpression. Assuming that cells grow at a constant exponential rate $\alpha$ during every cell cycle, one can then calculate for each generation the time ($T_z(g)$) that a cell needs to grow to reach $L_{stable}(g)$, which will allow the FtsZ binding to commence at mid-cell:

$$T_z(g) = \frac{1}{\alpha} \log \frac{L_{stable}(g)}{L_0(g)} \quad (2)$$

where $L_0(g)$ is the cell length at the start of generation $g$. Following the initiation of FtsZ binding, the cell then continues to grow for a fixed

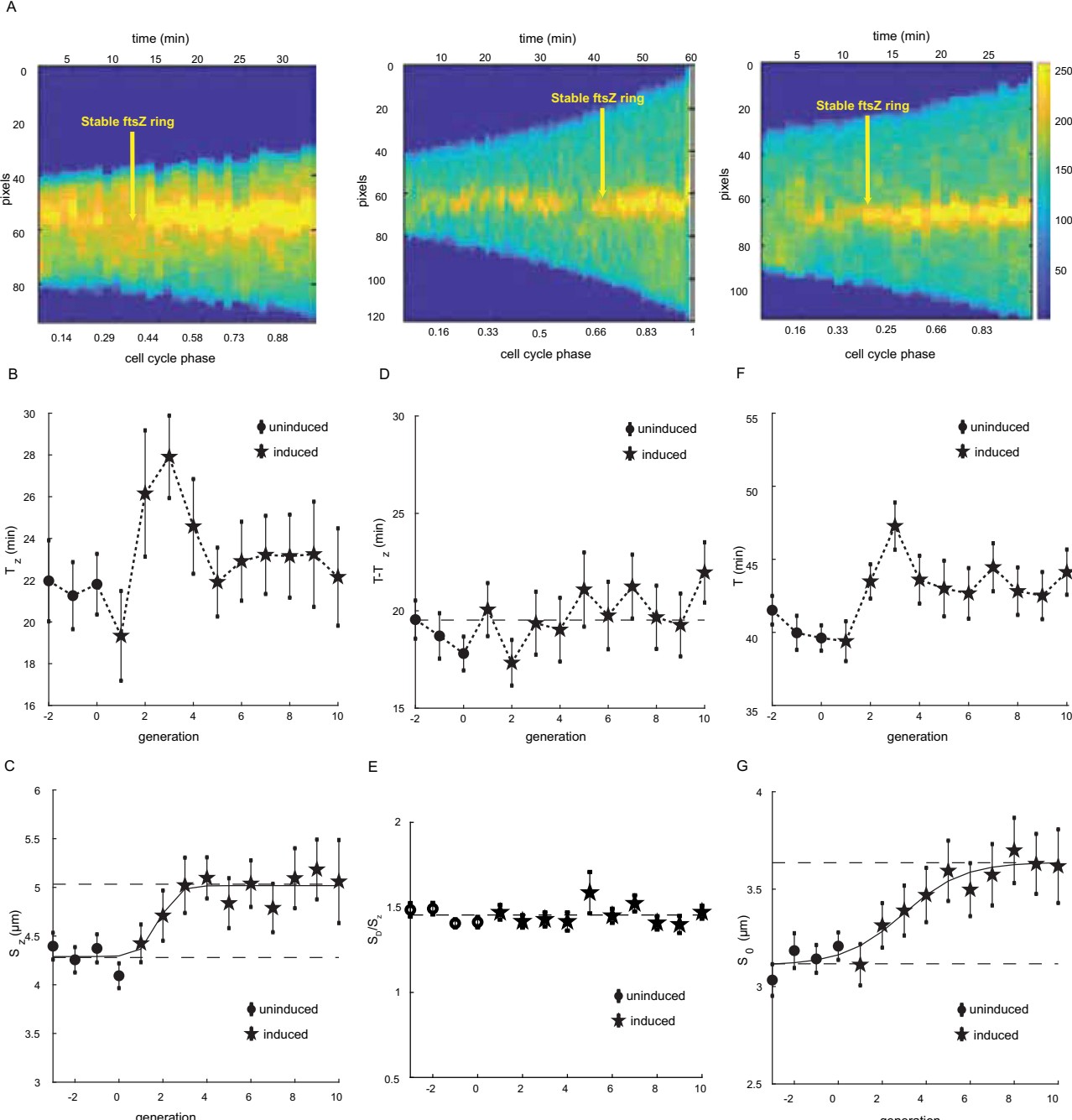

**Fig. 4 | Dynamics of FtsZ ring formation. A** Example heat maps depicting the FtsZ fluorescence intensity shows the FtsZ accumulation at the center of the cell during steady-state before inducing *minE* overexpression (left image), during the transition immediately after inducing *minE* overexpression (middle image), and after the cell reaches the new steady-state (right image). **B** Time to reach 65% of maximal FtsZ ring intensity increases transiently following overexpression of MinE proteins before decreasing back to its initial steady-state values. The 65% maximal FtsZ intensity was chosen as the threshold point, for which we measure the time needed to achieve a stable ring because, under uninduced *minE* conditions, we observe a clear change at that point in the rate of FtsZ accumulation within the ring. Note, however, that a choice of a lower or higher threshold only shifts our results without

changing the observed dynamics depicted here (see Fig. S5A, B). **C** Cell size measured at the time the FtsZ ring reaches 65% of the maximal intensity, i.e., the size corresponding to the time points in (**B**). **D** The time it takes for the cell to divide is measured from the point the FtsZ ring reaches 65% of its maximal value right before the division event. **E** The exponential increase in size ($S_D/S_z$) during the time intervals presented in (**D**). **F**, **G** Cell cycle duration (sum of (**B**) and (**D**)) and cell size at birth, respectively, measured before and during the induction of *minE* overexpression. Every point in the graphs presented in (**B**–**G**) represents the average calculated from 39 traces, and the error bars represent the standard error of the mean. For the complete size distributions at different time points, see Fig. S5C. Source data are provided as a Source Data file.

time τ as we see in Fig. 4D, which would lead to a cell length at the start of the subsequent generation:

$$L_0(g+1) = \frac{1}{2} \cdot L_{stable}(g) \cdot e^{\alpha\tau} = \frac{1}{2} \cdot L_0(g) \cdot e^{\alpha(T_z(g)+\tau)} \qquad (3)$$

where the factor 1/2 signifies the symmetric cell division event (Fig. S8). The results of this calculation for $T_z(g)$ and $L_0(g)$, are presented in Fig. 5C, D respectively. These results confirm that the FtsZ binding time, $T_z(g)$, exhibits transient dynamics, with the binding time increasing immediately following induction and gradually decreasing

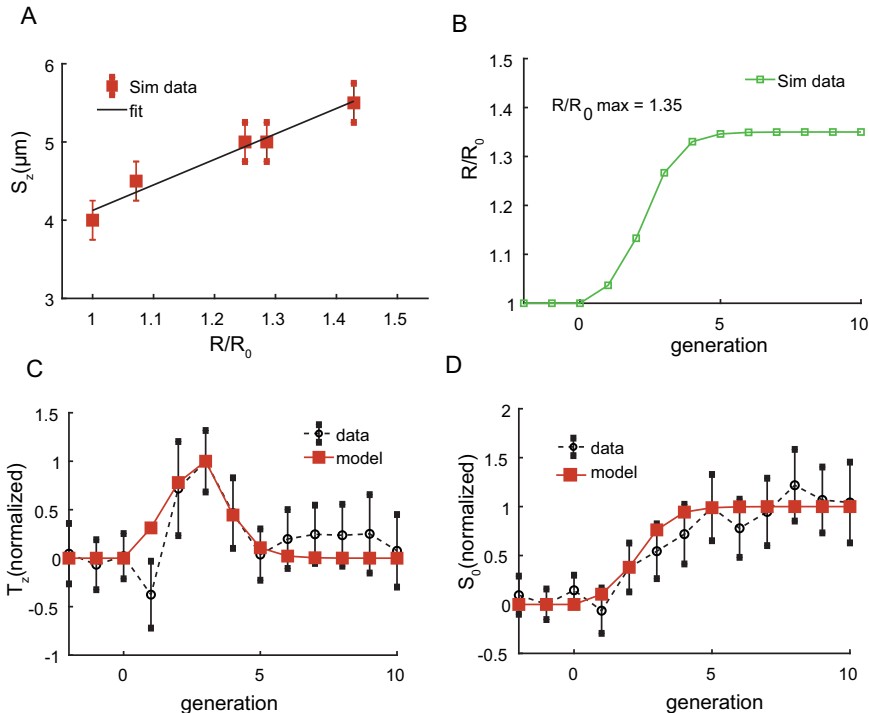

**Fig. 5 | Model analysis and simulation results. A** Simulations of the Min proteins oscillations were carried out for different MinE/MinD ratios and different cell sizes as described in "Materials and methods". The optimal size (defined in the main text) for each MinE/MinD ratio was obtained and plotted as a function of the change in the MinE/MinD ratio. Error bars depict the sampling resolution we used to determine the optimal cell size in the simulations, i.e., the difference between sizes sampled for each MinE/MinD ratio. The results show that the size at which a stable Z ring would form as a function of the change in the MinE/MinD ratio can be well-fitted by a linear function within a small range of change as a first-order approximation. This is in agreement with the experimental results presented in Fig. S7. Note that the increase in cell size scales linearly with the MinE/MinD ratio for almost a threefold change, in agreement with the population data in Fig. 2B. **B** The kinetics of MinE/MinD ratio (relative to the uninduced condition) following the induction of *minE* overexpression was evaluated as described in the main text. The changes in the time required for forming a stable FtsZ ring, and the expected cell size at birth, following the induction of *minE* overexpression, were calculated as described in the text and plotted (red squares) in (**C**) and (**D**), respectively, as a function of generation number, with generation zero being the induction initiation time. The dynamics of both $T_z$ and $S_0$ exhibit significant agreement with the experimental data (black circles) obtained from the single-cell measurements presented earlier in Fig. 4. Note that while the agreement between the dynamics observed in simulations and experiments is quantitative, only qualitative agreement is observed between them in the actual time and size measurements. This is due to the fact that the cell size that allows for stable FtsZ ring formation in the simulations depends on the kinetic parameters used, for which we do not have direct measurement and instead use the values from previous studies[63]. Source data are provided as a Source Data file.

as the *minE* overexpression approaches its new steady-state value, as we have observed experimentally (Fig. 4B). On the other hand, the cell length at the start of the cell cycle, $L_0(g)$, increases monotonically with the increase of *minE* expression (Figs. 4G and 5D).

### The sister cells test

One immediate testable prediction of the above-described results is that sister cells receiving different length fractions from their mother would add size that would be inversely proportional to the received fraction[3,58].

Although the distribution of Min proteins between sister cells can be asymmetric due to their pole-to-pole oscillations, it has been shown that, on average, it is very close to symmetric[66]. Since the concentrations of Min proteins that sister cells receive from their mother are similar on average, the sizes of both cells at the onset of the FtsZ ring formation should also be similar. This suggests that a cell born smaller than its sister will grow more than its larger sister before a stable FtsZ ring is formed (Fig. 6A). From that point on, each cell is expected to add the same length while the FtsZ ring is completed, and constriction can begin as shown in Fig. 4E. As a result, a shorter sister cell, will grow more than its larger sister during the cell cycle following division (Fig. 6B and Fig. S11). An additional contribution to the variation in the added size between sister cells can also result from differences in the growth and protein production

rates between sisters immediately after their separation, as we have recently reported[3,58].

## Discussion

In this study, we explored the role of the Min proteins in determining cell size in bacteria. Our results show that increasing the ratio of MinE/MinD protein concentrations results in an increase in the average cell size of *E. coli*. To further understand the mechanism underlying the observed increase in cell size, we performed single-cell experiments, in which we increased the expression of MinE relative to MinD proteins while tracking cell-size dynamics and FtsZ ring formation kinetics. These experiments revealed that modifying the balance of the Min proteins delays FtsZ binding to the membrane and the Z-ring formation initiation. We propose that this is due to the fact that the cell needs to reach a size that can accommodate a new dynamical pattern of the Min proteins (at the new concentrations) for which MinCD occupancy at mid-cell is low enough to allow a stable FtsZ ring formation. This, in turn, causes the cell to grow to a larger size before it divides and produces two daughter cells whose birth sizes are larger on average than the birth size of the mother. In agreement with this proposed mechanism, our measurements clearly show the increase in the average birth size of cells following the induction of *minE* overexpression. They further show that the increase in the birth size of daughter cells continues until the Min proteins reach their final steady-state

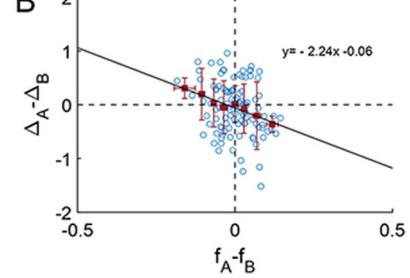

**Fig. 6 | The consequences of the Min proteins oscillation role in determining cell size to size homeostasis following asymmetric division. A** A cell born smaller during a division event will grow more before the FtsZ ring can start to assemble at the membrane. However, once the FtsZ ring assembly is stable, the cell will add a fixed size before it divides again. As a result, the smaller cell will add a larger volume during the cell cycle compared to its larger sister before they both divide again, as our experimental results confirm (**B**), which presents $\triangle_A - \triangle_B$ = (added length of sister cell A − added length of sister cell B) as a function of $f_A - f_B$ = (birth length of sister cell A − birth length of sister cell B)/(mother cell length). The graph contains 121 pairs of cells, and the error bars represent the standard deviation in each bin (for more data points, see Fig. S11). Source data are provided as a Source Data file.

concentrations following the induction of *minE* overexpression. On the other hand, the delay in the FtsZ ring formation appears to be transient and disappears once the cells reach the new steady-state size.

Several recent studies have convincingly argued that control of cell division involves two processes, DNA replication and replication-independent process[29,30,32–35]. In a recent study, it was shown that the onset of constriction in *E. coli* becomes independent of DNA replication termination under fast growth conditions[67]. This replication-independent process has been proposed to be the threshold accumulation of a protein, which could be the FtsZ protein, whose effect on the cell division timing has been demonstrated in multiple studies[19,29,38,68]. This proposal is also supported by studies in which increasing the cell width delayed the division process hinting at a need for a larger amount of FtsZ to complete the septal ring[29,32]. The results presented here support the view of complex multi-process control of the cell cycle and propose another layer of control that affects the initiation of septum formation. In the previous paradigm, the initiation of the Z-ring formation is fixed and independent of cell size. Here we show that the dynamics of the Min proteins determine the initiation time of the Z-ring formation by preventing the FtsZ binding to the membrane until their dynamical pattern allows it.

Cell size is an important factor for survival that organisms tune to their environmental needs. It has been long-established that bacterial cell size is strictly maintained within a narrow range while growing in a constant environment[69] but changes significantly in different environments[70–72]. Concerted control of cell size that integrates multiple checkpoints, as proposed here, can allow more precise control as each checkpoint can mitigate fluctuations in others. On the other hand, this can also provide additional plasticity that can help cells adjust their size to meet new environmental demands and adapt to survive new environmental challenges.

Finally, while our results agree with the predictions of a simple theoretical model of the Min oscillations, we point out that this agreement is qualitative only. To achieve a quantitative agreement, the model parameters should be determined experimentally. In addition, simulations of the Min oscillations should be carried out simultaneously with cell growth and protein production. Nevertheless, the findings presented here give new insights into cell-size control and variation in bacteria. They reveal that cell size is sensitive to the expression ratio of MinE/MinD, which may contribute to the "sloppy" nature of cell size control reported in a recent study[1], which has shown that different lineages in the same medium maintain homeostasis around variable average sizes. In addition, we remark that the cell size and FtsZ ring formation dynamics that we report here is the population average dynamics measured in response to a change in the Min proteins ratio beyond the physiological range expected under natural conditions. Therefore, to determine the contribution of the Min proteins to the variability in size homeostasis within a population, further experiments are needed, in which the ratio of the Min proteins could be measured at the single-cell level and in different conditions, and their correlation with cell size could be evaluated.

## Methods

### Population-level measurements of MinE/MinD effect on cell size
MG1655 bacteria were transformed with plasmids (gift of the Huang Lab, Stanford University) expressing MinE or MinD fused with mEos protein under the control of the arabinose inducible promoter $P_{araBAD}$[73]. This type of fluorescent protein fusion to MinE and MinD has been used in the past and was shown to be functional in vivo[44,47] and in vitro[50]. Cells were grown overnight in LB supplemented with the appropriate antibiotics at 32 °C while shaking at 240 rpm. The following morning, the culture was diluted 400-fold in the same medium and regrown for 1 h. The culture was then induced with arabinose and regrown at the same conditions for an additional 1.5 h. Different concentrations of arabinose were used to induce different levels of *minE–mEos* or *mEos–minD* expression in the cells. Following the 1.5 h of induction, samples were taken, and images of the cells were acquired using a Z1 inverted Zeiss microscope in phase contrast mode with a 100x objective in order to measure the cell lengths.

### Real-time RT-PCR
Bacteria were grown and induced with different levels of inducer concentration, as described above. The culture's $OD_{600nm}$ was monitored, and samples were collected when cultures reached an $OD_{600nm}$ ~ 0.3. The samples were then mixed with 2× RNA protect bacteria reagent (QIAGEN) and incubated at room temperature for 5 min. The mixture was centrifuged at 10,000 rpm for 10 min, the supernatant was discarded, and the pelleted bacteria were then stored at −80 °C. The following day, the RNA content of the cells was extracted using the RNeasy mini kit (QIAGEN) by carefully following the extraction protocol provided by the kit manufacturer. The extracted RNA was then used to estimate the ratio of *minE* and *minD* mRNA at different inducer concentrations by quantitative RT-PCR (See Supplementary Material and Methods), using the one-step QuantiTect SYBR green RT-PCR kit (QIAGEN), and the following primers (Integrated DNA Technologies) targeting *minE* and *minD*:

    minE_s: CGGCTGCAGATTATTGTTGC
    minE_as: TGCTCAAGCTGTACGGTTAC
    minD_s: GGTTTGGCCCAGAAGGGAA
    minD_as: TTAGCGTTGCATCGCCCTG

Measurement errors were determined by carrying out the quantitative RT-PCR on triplicate samples simultaneously.

## Single-cell level measurements of cell size and FtsZ ring formation dynamics

The bacterial strain KC376 (gift of the Huang Lab, Stanford University), which is a derivative of the MG1655 strain containing the fluorescent protein, mVenus, integrated within the *ftsZ* gene[61] was obtained from the Huang lab at Stanford University, and was transformed with two additional plasmids. The first, pZA3R-mcherry plasmid expressing the mCherry protein constitutively, was used to provide better detection of the cell boundaries in the microfluidic traps of the mother machine. The second was a plasmid expressing *minE–mEos*[74] under the control of $P_{araBAD}$.

The transformed KC376 cells were grown overnight at 32 °C in LB medium with the appropriate antibiotics with constant shaking at 240 rpm. The following morning the culture was diluted (1:400) in the same medium and grown until the culture reached an optical density ($OD_{600nm}$) ~ 0.1. Six milliliters of the culture were then collected into four 1.5 ml Eppendorf tubes and centrifuged at 1500 g for 6 min. The supernatant was discarded, and the concentrated cells were resuspended in 50 µl of fresh LB. Next, the cells were loaded into the mother machine and left to grow for 5 h without induction. Fresh LB medium was streamed through the device at a constant rate of 1 ml/h to supply nutrients to trapped cells and wash excess cells from the device. After 5 h of growth under uninduced conditions, the streamed LB medium was supplemented with arabinose to induce the expression of *minE–mEos* in the trapped cells. The resulting dynamics of cell-size change and FtsZ ring formation were then probed and mapped as described in the results section.

## Image acquisition and data analysis

For single cell analysis, cells growing in the microfluidic traps of the mother machine were imaged every 1 min in DIC and fluorescence modes using a Hamamatsu ORCA-flash 4.0 camera, mounted on a Nikon Eclipse Ti2 inverted microscope with a 100× objective at 32 °C, maintained using the microscope incubator (okolab, H201-1-T-UNIT-BL). Cell length was measured using the cell analysis software Oufti[75]. Custom MATLAB 2020a (MathWorks) programs were developed to analyze acquired data and calculate average statistics.

## Z-ring intensity measurement

The intensity of the FtsZ ring at the division septum of the cell was estimated by measuring the mean fluorescence intensity inside a 1 µm × 1 µm box at the center of the growing cells using image analysis software ImageJ 1.53k (Fig. 3Ainset). The mean intensity was corrected by subtracting the mean background illumination from all the images.

## Reporting summary

Further information on research design is available in the Nature Portfolio Reporting Summary linked to this article.

## Data availability

Source data are provided in this paper, and the data generated in this study have been deposited in the Zenodo database at https://doi.org/10.5281/zenodo.8282952. Source data are provided in this paper.

## Code availability

Supplementary software consisting of the 1-D simulation code used in this study is provided with this paper and has been deposited in the Zenodo database at https://doi.org/10.5281/zenodo.8282952.

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

## Acknowledgements

We thank the lab of K.C. Huang at Stanford University for the gift of bacterial strains used in this study. This study was supported by the National Science Foundation grant number Phy-2014116, and the USA – Israel Binational Science Foundation grant number 2016376 to H.S. Y.R. acknowledges support by grants from the Israel Science Foundation (178/16) and from the Israeli Centers for Research Excellence program of the Planning and Budgeting Committee (1902/12).

## Author contributions

H.V. carried out experiments and analyses, ran the simulations, and prepared figures. J.J.-T. carried out experiments and analyses. K.S. wrote the 1-D simulation code and ran the simulations. Y.R. oversaw the development of the 1-D simulations, advised on the procedures and analyses, and co-wrote the paper. H.S. designed the study, oversaw all the experiments, simulations, and analyses, and wrote the paper.

## Competing interests

The authors declare no competing interests.
