## [Peer Review File · Nature Communications]

Bacterial cell-size changes resulting from altering the relative expression of Min proteinsReviewer #1 (Remarks to the Author):

Vashistha and coworkers study the effect of Min proteins on the decision process leading to cell division in E. coli. While it is well known that these proteins form a spatio-temporal pattern that is able to localize the Z ring at mid-cell, much less is known on their effect on the decision to constrict and then divide. The authors show that increasing minE expression relative to minD, Z-ring formation is delayed and mean division size increases, and conclude that Min proteins are part of the decision to divide, as well as being a natural size sensor, by concentration.

The text is accessible and well explained. This study builds on previous work by the same authors (refs 11,27), but adds a mechanistic layer to these previous results, and connects them to other studies.

I think this study can make a nice addition to the current collective effort aiming to characterize cell-cycle progression in E. coli both mechanistically and at the single-cell level, in terms of cellular decisions.

However, it would be useful to elaborate on a few consequences of the main findings.

In particular, I think that a deeper study of single cell patterns and of the shift behavior after minE induction, as well as more comprehensive and more accurate literature comparisons are important to clarify the results.

Experimentally, testing another growth condition and looking at the single cell patterns for different concentrations of MinE could add important insight on the decision process leading to constriction (see below).

%%%

The more detailed comments below should clarify how I reached these conclusions:

%%%

Intro and results. The story is clear, also because the text is a bit redundant in some points (eg, the story of ftsZ accumulation at midcell is found at the beginning of many paragraphs). These repetitions are not problematic, but I find that the angle on the story is oversimplified. It is very likely FtsZ plays a primary role in setting adder correlations, but at the same time things are way more complex and there's a debate that the authors should discuss in the introduction and results. One important debate concerns limiting factors (Ref 17, Si et al, but also Witz et al PMID: 31710292, Micali et al PMID: 30417095 PMID: 30332654, Colin et al. PMID: 34612203, Tiruvadi Krishnan et al. PMID: 35320717, see also Kleckner et al. PMID: 30038602). The discussion should clarify how the authors' results connect to the debate on limiting factors / decision processes in cell division.

Tiruvadi Krishnan et al. PMID: 35320717 (TK) deserves a closer discussion. This article is cited, but only at a superficial level, and not in its final form, so possibly the authors did not see the published version of this work. It seems that in line with TK the authors find a timer period leading to constriction (but the distributions / correlations are not shown here), but it is not clear how the division in subperiods used in Fig 4 relates to the one used by TK. In addition, TK advocates that the onset of constriction is coupled to replication-related processes in slow growth conditions, while there is replication-segregation and FtsZ-related regulation work on competing time scales. Hence, I expect that MinE titration could work differently depending on the growth conditions. This hypothesis could be tested experimentally.

Another central question is whether in unperturbed conditions Min proteins play a role, or the early formation of a stable ring makes other processes more relevant. In other words the action of minE overexpression could be analogous to the delay of cell division caused by A22 (see Colin et al. PMID: 34612203 Zheng et al 2016 PMID: 27956612)

It would be nice to parallel the results more precisely. Experiments looking at single cell patterns at different target expression levels of MinE could be important to disentangle this question.

Can the authors clarify how their model leads to added correlations and what are the single-cell predictions for the two subperiods at steady state?

What are the experimental observations and the model prediction for the interdivision added plot (or equivalent), and for the Tz and T-Tz subperiods?

Also, the authors use the model to study the shift (Fig 5). Panlilio et al. PMID: 33931503 use a FtsZ-like accumulator model to describe the nutrient upshift. This model leads to multiple time-scales behavior similarly to Fig 5. The shift behavior can also be compared to the model used in Buke et al. PMID: 34990598, also describing a shift to a perturbation where ppGpp levels change (affecting cell size in a direct way, in a similar fashion as observed in Fig 4). Can the authors clarify the similarities and differences? Does their model effectively work as an accumulator triggering division?

In connection with this point, Fig 5 seems to lack a model/data comparison.

The vertical asymptote in Fig 2B may suggest that cell size keeps increasing with minE concentration even if the ratio with MinD expression, R remains constant (hence a role of minE beyond the ratio). Is this reasoning wrong? Can the authors clarify why?

Figure S2. Does the size distribution under this perturbation obey the collapse properties observed in other studies (e.g. Kennard et al. PMID: 26871102, Salman et al PMID: 23003996, but also refs 34 and 16) or does the perturbation break it?

The results appear clear and robust but data visualization could be improved: (i) in Fig 1 the intercept is the same (size of the unperturbed cells), but the style of the y axes in the two panels is a bit misleading, and the two curves cannot be compared quantitatively (which is crucial to some conclusions). (ii) in Fig 2 there are strange extra ticks on the y axis (iii) in Fig 4 a label of the X axis is missing in panel F (but not in panels BD) The legend text is small, and a clearer label for the induction time could help. Panel A could also be plotted as a function of cell cycle phase (it should look like Fig 3C and 3D panels), (iv) in Fig5 the legend is missing, it is not clear what the symbols vs lines represent and the caption is overly concise.

p3 typo (extra TO ?) "the timing of the FtsZ ring placement TO coincides with the"

Reviewer #2 (Remarks to the Author):

The manuscript by Vashistha et al. reports how MinE-mEos upregulation affects cell length in *E. coli*. The authors find that upregulation increases cell length but at higher inducer levels, the length increase stops. The manuscript's main portion focuses on how cell size changes in time since the upregulation of MinE-mEos. The authors find the length increase already during the first division after induction and cell length reaches a new steady state in about 5-6 generations. Interestingly, they find that the cell cycle duration shows overshoot albeit in a single datapoint at generation 3 (Fig. 4F). The authors also find that the timing for the "stable" Z-ring also shows overshoot, but it remains unclear what this timing exactly signifies. The authors then build a cell cycle model to explain such behavior. The model relies on a simple model for the Min oscillations, whose validity remains questionable. Also, the sizer-like cell cycle model that the authors propose has been refuted by a large body of literature.

The overshoot behavior that the authors report is interesting but not unexpected. Any significant perturbation FtsZ concentration or its ability to form the Z-ring can be expected to lead to such behavior. For example, Si et al. *Curr. Biol.* 2019 observed such behavior when they down-

regulated FtsZ level (they performed periodic perturbation instead of step-like perturbation). Furthermore, it is not feasible that an increase of $[\text{MinE}]/[\text{MinD}]$ by about 3x from its mean value would occur in wild-type cells where these two proteins are expressed from a single operon. The concluding statement in the abstract that "These results highlight the contribution of Min proteins to cell size control, which can account for part of the size fluctuations observed in bacterial populations, and can clarify how size difference acquired during asymmetric cell division is offset" is therefore not backed up by the data. This is also acknowledged by the last sentence in the manuscript "Further experiments are needed to address the contribution of the Min proteins to the variability in size homeostasis within a population, in which the ratio of the Min proteins could be measured at the single-cell level and in different conditions, and their correlation with cell size could be evaluated." Unless these measurements are performed the presented conclusions remain sketchy.

Main points of criticism:

1) The authors make several assumptions about Min oscillations (such as described at the end of page 5 and beginning of page 6 but also elsewhere). It is unclear if these assumptions are correct. Instead of speculations, they should present experimental data to show how the distribution of MinD changes at different concentrations of MinE and different cell lengths. The predictions of their 1D model for these oscillations are, at best, qualitative. Furthermore, the model seems to show robust Min oscillations in all conditions (Fig. S5). At the same time, the authors claim in several passages that there is no "regular Min proteins dynamical pattern early in the cell cycle." (this quote is from page 5, but it appears on several other occasions too). The authors should also present evidence for the lack of regular Min patterns in the cell cycle and, if needed, adjust their model to account for the lack of these patterns. Some discussion quantification on conditions when the Min oscillations are lacking is also warranted.

2) The studied $[\text{MinE}]/[\text{MinD}]$ ratio is far out of range to what can be expected in normal growth conditions. "These results highlight the contribution of Min proteins to cell size control, which can account for part of the size fluctuations observed in bacterial populations, and can clarify how size difference acquired during asymmetric cell division is offset" is therefore not backed up by the data.

3) Using RT-PCR for protein ratio is not justified except perhaps comparing one induction level to another. RT-PCR does not measure protein concentration. Using RT-PCR based $[\text{MinE}]/[\text{MinD}]$ ratio in the model for the Min oscillation is not valid. Furthermore, the model for the Min oscillations is poorly described. It is unclear what rate constants and the absolute concentrations for MinD and MinE are used in the model.

4) Using MinE-mEos instead of regular MinE in upregulation experiments (the only experiment performed) is a poor choice. It is unclear if the fusion behaves the same way as the wild type protein. Some controls regular MinE need to be performed, but a better choice would be to redo all the measurements since there is effectively only a single measurement. Also, mEos fluorescence overlaps with mVenus one (in a non-photoactivated state), making the choice of using this construct even more questionable.

5) It is unclear why 65% of FtsZ intensity at the Z-ring is considered the critical threshold for cell division in *E. coli*. How would the results (T_z , S_z etc.) change if some other level is used? In any case, some justification for this threshold is needed.

6) The rationale for presenting new data (Fig. 6) at the end of the Summary and Discussion section is unclear. It would be more meaningful to present it at the end of the Results section.

Minor points of criticism:

1) Introduction: "The variations, though, are confined to a restricted range specific to the property being quantified ..." Not clear what point the authors aim to make. I do not see any information conveyed by this sentence.

2) Introduction: "Further support was obtained also by demonstrating that reducing the amount of

FtsZ or increasing its degradation rate delayed cell division^{30,31}."

It does not seem that references 30, 31 are the correct ones to support this statement.

Contrary to ref 17, ref 30 states that "Our results suggest that septation is not triggered by a fixed number of newly synthesized FtsZ molecules per cell." Also, "The number of FtsZ molecules per cell can be reduced threefold without affecting the division rate significantly."

Ref 31 states, "These results support a model in which the timing of FtsZ assembly is governed primarily through cell cycle-dependent changes in FtsZ polymerization kinetics and not simply via oscillations in the intracellular concentration of FtsZ." This is also not in support of ref 17.

None of these two works discuss degradation of FtsZ.

3) Introduction: "It has been demonstrated in many studies that the placement of FtsZ in the cell is determined by oscillatory dynamics of membrane-associated proteins, collectively termed the Min system³²⁻⁴⁰." There are additional systems also involved in this placement and these should be mentioned. Furthermore, it should be mentioned that the Min system is not essential for the midcell placement of the Z-ring, although it prevents divisions at cell poles.

4) Introduction: "This in turn allows the stable binding of FtsZ to the membrane". Unclear what "stable" means here. The residency times of FtsZ molecules on the cell membrane are about 10 seconds.

5) Introduction: "These results demonstrate the crucial effect of the fluctuations in Min proteins on the FtsZ ring formation". Unclear what "crucial" means. The cells that lack the Min system still form Z-rings and divide, yet some divisions produce minicells.

6) Results: "We argue that this effect is mediated by the need to have an appropriate cell size to allow for a regular Min proteins dynamical pattern early in the cell cycle." Measurements should show this.

7) Fig. 2B : Not clear what the error bars show.

8) Results: "Thus, the cell size at birth is one that allows the existing concentrations of Min proteins in the cell to produce stable patterns early in the cell cycle." The authors should show these patterns.

9) Results: "therefore cell division would be delayed until the cell reaches a new length that would allow a regular stable Min oscillation with the new concentrations."

10) Results: "Once FtsZ binds stably to the membrane, the ring starts to form, and the cell will continue to grow until the FtsZ ring is completed and the cell is ready to divide." Please clarify what "stably" means. The FtsZ in the Z-ring turns over in the time scale of 10 seconds (Stricker & Erickson PNAS 2002). Please also clarify what means "the FtsZ ring is completed". In what sense the FtsZ ring is completed?

11) "Our observations show that immediately following the induction of minE-mEos expression, cells become longer". Unclear what is meant. I assume cells continue to grow.

12) Fig. 3B : "Final cell size" - I would call it cell length at division. "Elongating" - after induction (are there cells that are not elongating?). What do the error bars show?

13) Fig. 3C-D: FtsZ intensity at midcell should decrease at the end of the cell cycle, but this decrease is missing, indicating some problems in the analysis.

14) ftsZ -> FtsZ. What does "stable" mean?

15) Fig. 4B-G: Why are the error bars based on bootstrapping? The reader deserves to see std or std errors based on conventional calculation.

16) Fig. 4: Referring "normal" before induction can be misleading. Potentially $[MinE]/[MinD]$ is significantly higher than in WT cells. The authors should mention what is $[MinE]/[MinD]$ before

induction. The applicability of determining this ratio using RT-PCR should also be discussed.

17) Summary and Discussions: "These experiments revealed that disturbing the balance of the Min proteins delays FtsZ binding to the membrane and the Z-ring formation initiation until the cell reaches a size that allows for a stable regular dynamical pattern of the Min proteins with the new concentrations" I cannot find these experiments described in the manuscript.

18) Summary and Discussions: "More specifically, since the concentrations of Min proteins that sister cells receive from their mother are similar, their sizes at the onset of the FtsZ ring formation should be similar." Not clear if this assumption is correct, taking that the Min system shows no "stable regular dynamic" pattern at early stages of the cell cycle. Furthermore, the assumptions the authors make about FtsZ distribution between daughter cells need to be explained.

19) The authors' model is effectively a sizer-model with a critical cell cycle checkpoint when the FtsZ at the Z-ring reaches 65% of its maximum level. Interesting, but the evidence comes from a single poorly defined measurement (MinE-mEos upregulation). There is a large body of literature based on much more explicit measurements that refute sizer models.

20) Fig. 6B: Based on large error bars, it is unclear if there is a significant trend. The authors present no statistical test. Also, there is no explanation of what the error bars show, how many cells were analyzed, what conditions the experiments were done, etc.

Dear Editor,

We would like to thank the reviewers for their efforts in assessing our manuscript and for their valuable feedback, which we address in detail below. We believe that after addressing the reviewers' concerns, the manuscript is now clearer, and the results are more convincing. All the changes we made in the main text as well as the supporting material are labeled in red for easier tracking. We hope that the reviewers will agree with our assessment and find the manuscript now suitable for publication.

On behalf of all authors,

Hanna Salman

Response to reviewers' criticism:

Reviewer #1 (Remarks to the Author):

Vashistha and coworkers study the effect of Min proteins on the decision process leading to cell division in E. coli. While it is well known that these proteins form a spatio-temporal pattern that is able to localize the Z ring at mid-cell, much less is known on their effect on the decision to constrict and then divide. The authors show that increasing minE expression relative to minD, Z-ring formation is delayed and mean division size increases, and conclude that Min proteins are part of the decision to divide, as well as being a natural size sensor, by concentration.

The text is accessible and well explained. This study builds on previous work by the same authors (refs 11,27), but adds a mechanistic layer to these previous results, and connects them to other studies. I think this study can make a nice addition to the current collective effort aiming to characterize cell-cycle progression in E. coli both mechanistically and at the single-cell level, in terms of cellular decisions.

We agree with the reviewer's summary of the main point of our manuscript, and we appreciate their evaluation of our work as a nice addition to the field, which improves our mechanistic understanding of cell size control.

However, it would be useful to elaborate on a few consequences of the main findings.

In particular, I think that a deeper study of single cell patterns and of the shift behavior after minE induction, as well as more comprehensive and more accurate literature comparisons are important to clarify the results.

Experimentally, testing another growth condition and looking at the single cell patterns for different concentrations of MinE could add important insight on the decision process leading to constriction (see below).

We appreciate this critique, and we now added more experimental results of single-cell measurements in a different medium as well as with different induction levels of minE, which provide further support to our previous results and conclusions. These results are presented in the new supporting figure Fig. S5 and as an inset to Fig. 6A. The results are also discussed in the main text before Fig.4.

The more detailed comments below should clarify how I reached these conclusions:

Intro and results. The story is clear, also because the text is a bit redundant in some points (eg, the story of ftsZ accumulation at midcell is found at the beginning of many paragraphs). These repetitions are not problematic, but I find that the angle on the story is oversimplified. It is very likely FtsZ plays a primary role in setting adder correlations, but at the same time things are way more complex and there's a debate that the authors should discuss in the introduction and results. One important debate concerns limiting factors (Ref 17, Si et al, but also Witz et al PMID: 31710292, Micali et al PMID: 30417095 PMID: 30332654, Colin et al. PMID: 34612203, Tiruvadi Krishnan et al. PMID: 35320717, see also Kleckner et al. PMID: 30038602). The discussion should clarify how the authors' results connect to the debate on limiting factors / decision processes in cell division.

Tiruvadi Krishnan et al. PMID: 35320717 (TK) deserves a closer discussion. This article is cited, but only at a superficial level, and not in its final form, so possibly the authors did not see the published version of this work. It seems that in line with TK the authors find a timer period leading to constriction (but the distributions / correlations are not shown here), but it is not clear how the division in subperiods used in Fig 4 relates to the one used by TK. In addition, TK advocates that the onset of constriction is coupled to replication-related processes in slow growth conditions, while there is replication-segregation and FtsZ-related regulation work on competing time scales. Hence, I expect that MinE titration could work differently depending on the growth conditions. This hypothesis could be tested experimentally.

We agree with the reviewer assessment, and we thank them for the valuable input and suggestions of additional information. We have reviewed these papers and discuss their relationship to our work in the second paragraph of the introduction and the paragraph before last of the discussion. In short, we think that our new results are in line with the previous findings. We specially discuss the importance of DNA replication in determining division events and we explain that based on previous findings, the replication becomes a limiting factor in slow growth conditions, as the reviewer also hints at the end of their comment. Therefore, we make it clear now that our study focuses on the effect of the Min proteins on cell size in fast growth conditions, where it has been demonstrated in the past that FtsZ threshold accumulation is a limiting factor for division. We agree with the reviewer that testing the effect of Min proteins in other growth conditions such as slow growth, will provide very important insight into the problem, but this is beyond the scope of the current study. We hope that with the additional tests that we now provide, the reviewer will still find that the study offers interesting results worthy of publication on their own.

Another central question is whether in unperturbed conditions Min proteins play a role, or the early formation of a stable ring makes other processes more relevant. In other words the action of minE overexpression could be analogous to the delay of cell division caused by A22 (see Colin et al. PMID: 34612203 Zheng et al 2016 PMID: 27956612)

It would be nice to parallel the results more precisely. Experiments looking at single cell patterns at different target expression levels of MinE could be important to disentangle this question.

This is another interesting point that the reviewer brings up. We think that our results here agree with both studies the reviewer mentions and suggest that the replication-independent mechanism, which Colin et al propose could be indeed the FtsZ accumulation to a threshold. In both of these studies, it is observed that increasing the width of the cell by using A22 increases the time interval from replication

termination to cell division, and reduces the correlation between cell division and replication termination. As clearly suggested in Colin et al, the delay of the division event could be due to the need for more FtsZ to complete a larger septum ring. In Zheng et al, they directly test the effect of FtsZ reduced expression and see the same effect on division time. The difference Zheng et al observe is that in the case of A22 the cell length doesn't seem to increase, while when FtsZ expression is reduced it does. This could be due to the fact that a wider cell would need more components to grow, i.e. a wider cell needs to increase a larger cell area than a narrower one. And it has been shown in another recent study (Oldewurtel et al PNAS 2021), that the cell surface grows in proportion to the biomass.

We discuss this point more in the second paragraph of the introduction and the paragraph before last of the discussion.

As for the suggestion of experimentally visualizing the Min proteins patterns, we are unable to carry out these measurements here mainly because such experiments would require rapid fluorescence imaging of cells, since the oscillation period is on the order of 10s of seconds. This rapid imaging damages the cells and as a result, we cannot follow the cell size changes over a time period of generations. In fact, we had to optimize our imaging rate (image/minute) so that we can track a total of 10 generations after induction in order to measure how the cell length changes and stabilizes to a new value. In addition, such measurements have been carried out in multiple previous studies, including an in vitro study by the Schwille group in Science 2008, which clearly showed how the pattern wavelength changes with the concentration of MinE, when MinD concentration is fixed.

That's why we had to settle with comparing our measurements to the prediction of simulations, which have been repeatedly confirmed in the past.

Can the authors clarify how their model leads to adder correlations and what are the single-cell predictions for the two subperiods at steady state?

Our proposed model of cell size control does not change the adder correlation or the predictions for the cell-cycle subperiods. It only suggests that some of the variation in cell size we observe could be a result of fluctuations in the Min proteins abundance in the cells, which affect the initiation time of the Z-ring formation. This is supported by our examination of the differences in size between sister cells, which shows that, on average, the smaller sister grows more during the first cell-cycle after separation. However, this too, does not change the adder correlation, because between the two sisters, on average (which is the source of the adder correlation), both sisters add a fixed size (Fig. S7).

What are the experimental observations and the model prediction for the interdivision adder plot (or equivalent), and for the Tz and T-Tz subperiods?

As mentioned above, the predictions for the adder plot are not altered by our results. In fact, Fig. 4E shows that if we measure the cell size added from the point, where a stable Z-ring is formed until the end of the cell cycle, we see that this size is constant. This is consistent with the view that under fast growth conditions (where replication is not a limiting factor, and division is primarily determined by the replication-independent process, i.e. threshold accumulation of FtsZ) the cell accumulates constant size as it is building the FtsZ-ring to the threshold needed for constriction initiation. Our results show that this is true even during perturbations, and the only thing that changes is the size accumulated prior to reaching stable FtsZ ring. This is true as well for the time that elapses from the instant the Z-ring is stably

formed until the end of the cell cycle, i.e., T-Tz (Fig. 4D). Again, the only thing that changes here is the time to form a stable FtsZ ring (Tz, Fig. 4B).

Our model predictions for Tz are presented in Fig. 6C. However, we do not have any prediction for T-Tz, since we assume constant growth for constant time after forming a stable Z-ring, based on our experimental observation in Fig. 4D. In other words, we use our experimental observation for T-Tz, and test whether this, together with what we already know about the Min oscillations, can explain the temporal dynamics of the birth size of the cell.

We stress again that we are not proposing a new cell-size control mechanism. We are adding another layer, which could account for part of the fluctuations observed, and explain the “sloppy” nature of the control mechanism already established. This layer does not alter the experimental observations on average at the population level, nor does it change the predictions from previous models that explain the added correlations.

Also, the authors use the model to study the shift (Fig 5). Panlilio et al. PMID: 33931503 use a FtsZ-like accumulator model to describe the nutrient upshift. This model leads to multiple time-scales behavior similarly to Fig 5. The shift behavior can also be compared to the model used in Buke et al. PMID: 34990598, also describing a shift to a perturbation where ppGpp levels change (affecting cell size in a direct way, in a similar fashion as observed in Fig 4). Can the authors clarify the similarities and differences? Does their model effectively work as an accumulator triggering division?

We do not think that our observations here are related to these previous observations. In the previous studies, the perturbations used, i.e. nutrient upshift and ppGpp level, are global factors that affect cells in multiple ways, whereas here, we only change the expression level of one specific protein.

However, as we stated previously, our model does work as an accumulator triggering division. That accumulation is of the FtsZ proteins, which need to reach a certain threshold in the Z-ring to initiate constriction. The only change we introduce is that the start of that accumulation can shift based on the concentrations of the Min proteins and the cell size at birth.

In connection with this point, Fig 5 seems to lack a model/data comparison.

We have now added this comparison and the figure now is Fig. 6.

The vertical asymptote in Fig 2B may suggest that cell size keeps increasing with minE concentration even if the ratio with MinD expression, R , remains constant (hence a role of minE beyond the ratio). Is this reasoning wrong? Can the authors clarify why?

In the measurements we present, R is different in all measurements. What Figure 2B shows is that small changes to R/R_0 past a certain point cause significant change in cell size. This might hint that there is a certain threshold for R , past which the cell will filament as the reviewer suggests. However, as we clarify now in the figure caption, from our measurements we cannot predict what would happen if we kept increasing the expression of MinE. The expression in our experiments is capped at the fully induced state of the plasmid, which we show in Figure 2A. Past that point, it is possible that the length saturates and stops increasing, or that the cell filaments and its length becomes uncontrollable. Based on the available data at this point we cannot claim that minE could have a role beyond the ratio.

Figure S2. Does the size distribution under this perturbation obey the collapse properties observed in other studies (e.g. Kennard et al. PMID: 26871102, Salman et al PMID: 23003996, but also refs 34 and 16) or does the perturbation break it?

We have now included in the same figure the scaled distributions which show collapse as in our previous publication Salman et al PMID: 23003996.

The results appear clear and robust but data visualization could be improved:

- (i) in Fig 1 the intercept is the same (size of the unperturbed cells), but the style of the y axes in the two panels is a bit misleading, and the two curves cannot be compared quantitatively (which is crucial to some conclusions).
We appreciate this observation. We now use the same scale in both graphs.
- (ii) in Fig 2 there are strange extra ticks on the y axis
Extra ticks have been removed.
- (iii) in Fig 4 a label of the X axis is missing in panel F (but not in panels BD) The legend text is small, and a clearer label for the induction time could help. Panel A could also be plotted as a function of cell cycle phase (it should look like Fig 3C and 3D panels),
The label of x-axis in 4F have been added, and we have added the cycle phase as the x-axis, with the actual time on the top of each panel.
- (iv) in Fig5 the legend is missing, it is not clear what the symbols vs lines represent and the caption is overly concise.
Legends added.

p3 typo (extra TO ?) “the timing of the FtsZ ring placement TO coincides with the”
Corrected.

Reviewer #2 (Remarks to the Author):

The manuscript by Vashistha et al. reports how MinE-mEos upregulation affects cell length in E. coli. The authors find that upregulation increases cell length but at higher inducer levels, the length increase stops. The manuscript's main portion focuses on how cell size changes in time since the upregulation of MinE-mEos. The authors find the length increase already during the first division after induction and cell length reaches a new steady state in about 5-6 generations. Interestingly, they find that the cell cycle duration shows overshoot albeit in a single datapoint at generation 3 (Fig. 4F). The authors also find that the timing for the "stable" Z-ring also shows overshoot, but it remains unclear what this timing exactly signifies. The authors then build a cell cycle model to explain such behavior. The model relies on a simple model for the Min oscillations, whose validity remains questionable. Also, the sizer-like cell cycle model that the authors propose has been refuted by a large body of literature.

We would like to thank the reviewer for their time and effort in evaluating our work, and for the criticism and feedback that has allowed us to significantly strengthen and improve our study. We carried out additional experiments in order to address some of the concerns raised by the reviewer and modified the

manuscript in order to clarify issues that may have led to misunderstanding of the implications of the results we present here.

More specifically, we would like to stress that we do not propose a sizer-like cell cycle model. We contend that from the point of forming a “stable” Z-ring until the end of the cell cycle, the cells add a constant size as have been demonstrated in many previous studies (Fig. 4D). Our results only add another factor to the cell-size control, which takes place early in the cell-cycle prior to the Z-ring formation. This factor, the Min system oscillations, can lead to variation in the added cell size, with the population average still exhibiting an adder-like cell-size control. The fact that sister cells can add different sizes based on their initial relative size does not lead to a sizer-like control mechanism, because when averaged it will appear as if each cell added a fixed size (see Fig. S7). Note that the adder phenomenon is a population average phenomenon, with large variability around the average behavior. Here, we argue that part of that variability could be a result of the role the Min proteins play prior to forming the Z-ring. We demonstrate this by altering the ratio of the Min proteins somewhat significantly in order to produce a detectable effect.

Second, the model we used to simulate the Min oscillations is the simplest model that can produce oscillations similar to those observed experimentally, and similar models have been used in many previous studies. Here, we show that using this simplest description of the interactions among the Min proteins (note for example that we do not consider MinC at all) we are able to again explain the observed dynamics of cell size change quantitatively (see new comparison in Fig. 6).

As to the timing of the “stable” Z-ring formation, we now clarify at the end of the introduction what we mean by “stable”, which refers to a Z-ring that does not degrade and continuously grows until the end of the cell cycle or the division event. We also show that the behavior we observe, i.e., the increase and decrease in the time required to achieve a stable ring, does not depend on where we choose to identify the ring as stable, e.g., at 65% or less (more) of the maximum intensity (Fig.S4).

The overshoot behavior that the authors report is interesting but not unexpected. Any significant perturbation FtsZ concentration or its ability to form the Z-ring can be expected to lead to such behavior. For example, Si et al. Curr. Biol. 2019 observed such behavior when they down-regulated FtsZ level (they performed periodic perturbation instead of step-like perturbation).

In Si et al, the authors showed that regulating the expression of FtsZ can affect the cell size because the construction of the Z-ring is affected by that. Here on the other hand, we show that without interfering with FtsZ expression, delaying the initiation of the Z-ring formation can affect cell size as well. We agree with the reviewer though that our results are related to those of Si et al, as we state in the manuscript, except that we add another factor that contributes to the cell size by influencing the initiation of the Z-ring formation. While this might not be unexpected, as the reviewer states, we are not aware of any other study that showed this effect previously.

Furthermore, it is not feasible that an increase of [MinE]/[MinD] by about 3x from its mean value would occur in wild-type cells where these two proteins are expressed from a single operon. The concluding statement in the abstract that "These results highlight the contribution of Min proteins to cell size control, which can account for part of the size fluctuations observed in bacterial populations, and can clarify how size difference acquired during asymmetric cell division is offset" is therefore not backed up by the data. This is also acknowledged by the last sentence in the manuscript "Further experiments are

needed to address the contribution of the Min proteins to the variability in size homeostasis within a population, in which the ratio of the Min proteins could be measured at the single-cell level and in different conditions, and their correlation with cell size could be evaluated." Unless these measurements are performed the presented conclusions remain sketchy.

We appreciate the reviewer criticism that we did not provide enough data to support such a strong claim as we made. We have now added new single-cell measurements under a different growth condition as well as under different induction levels of minE (detailed below), which provide further support to our claims. We show that the dynamics of cell size change and cell cycle duration following the induction is independent of the level of minE induction (Fig. S5). We further show that the changes in size and cycle duration scale with the MinE/MinD ratio (Fig. S5 insets). This allows us to extrapolate our results to lower levels of variation. This is within the norms of biological experimental research, where a system is tested in extreme conditions out of its natural range to produce a measurable effect beyond the measurement's errors. We trust that the reviewer will agree that this approach does not reduce the value of the experiments and can be used to draw conclusions to what might happen under natural conditions, which can be tested by other means in future studies. For example, in the study the reviewer cites in their note (Si et al), the authors exaggerated the expression fluctuations of the FtsZ to measure how it affects the cell size.

We hope that now the reviewer will find the new manuscript suitable for publication.

Main points of criticism:

1) The authors make several assumptions about Min oscillations (such as described at the end of page 5 and beginning of page 6 but also elsewhere). It is unclear if these assumptions are correct. Instead of speculations, they should present experimental data to show how the distribution of MinD changes at different concentrations of MinE and different cell lengths. The predictions of their 1D model for these oscillations are, at best, qualitative. Furthermore, the model seems to show robust Min oscillations in all conditions (Fig. S5). At the same time, the authors claim in several passages that there is no "regular Min proteins dynamical pattern early in the cell cycle." (this quote is from page 5, but it appears on several other occasions too). The authors should also present evidence for the lack of regular Min patterns in the cell cycle and, if needed, adjust their model to account for the lack of these patterns. Some discussion quantification on conditions when the Min oscillations are lacking is also warranted.

We appreciate this important point raised by the reviewer. We have indeed mischaracterized the Min oscillations and their regular patterns. We now explain in the first paragraph of the section "Experimental results confirm model predictions" that a regular oscillatory pattern always exists; however, it changes with the conditions, and as a result it can sweep the cell in a different manner and different frequency that would not allow the FtsZ ring to accumulate uninterrupted. This in turn will cause the decomposition of the FtsZ ring until a new pattern is achieved that would allow the stable accumulation of FtsZ and the completion of the septum ring as demonstrated in Fig. 4A middle image. We also included 2 new simulation movies (Videos SV2 and SV3) that demonstrate this behavior.

2) The studied [MinE]/[MinD] ratio is far out of range to what can be expected in normal growth

conditions. "These results highlight the contribution of Min proteins to cell size control, which can account for part of the size fluctuations observed in bacterial populations, and can clarify how size difference acquired during asymmetric cell division is offset" is therefore not backed up by the data.

The reviewer correctly states that the MinE/MinD ratio is far out of the range of what can be expected in normal growth. However, as we explained above, it is a common practice in experimental research to test the behavior of a system under extreme conditions, and use the results to make conclusions about the natural conditions. We have now strengthened our claims using new single-cell measurements with different levels of minE induction. These experiments show that the behavior of T (cell cycle time) is the same under all induction levels (Fig. S5), which implies that the time that the induced plasmid needs to reach the new steady state expression is the same regardless of the induction level, as we show in Fig. S9. In addition, we show that the final cell size linearly scales with the MinE/MinD ratio (Fig. 6a), which is in agreement with the model prediction in this range of expression and supports our claim that we can extend our conclusions to natural conditions.

3) Using RT-PCR for protein ratio is not justified except perhaps comparing one induction level to another. RT-PCR does not measure protein concentration. Using RT-PCR based [MinE]/[MinD] ratio in the model for the Min oscillation is not valid. Furthermore, the model for the Min oscillations is poorly described. It is unclear what rate constants and the absolute concentrations for MinD and MinE are used in the model.

The reviewer is correct in writing that RT-PCR does not measure protein concentration. However, as we explain in the third paragraph of the first results subsection, "...that while the mRNAs ratio might not reflect the proteins ratio 1:1, the proteins ratio will be linearly proportional to the mRNAs ratio. Therefore, we can accurately determine how this ratio changes relative to a fixed reference condition, which we take here to be the uninduced condition." In other words, we do not measure the mRNA ratio at each condition and use it as the protein ratio, but rather we measure how that ratio of the mRNAs of both genes changes relative to their ratio at another condition. So, if for example, the proteins ratio MinE/MinD in one condition is 5, while the mRNAs ratio minE/minD is 2, then we can conclude that if the mRNAs ratio becomes 4, the proteins ratio will be 10 and therefore the relative change remains the same.

More importantly, the exact quantitative relationship between MinE/MinD and cell size is not a main result of the study. The main point is that a change in this ratio can lead to cell size change via its effect on the timing of the Z-ring formation.

As for the rate constant and concentrations, they can be found in the supplementary information provided, where we include the simulation code. The rates and concentrations we use as reference are the same used in the original simulations, on which our simulations were based.

4) Using MinE-mEos instead of regular MinE in upregulation experiments (the only experiment performed) is a poor choice. It is unclear if the fusion behaves the same way as the wild type protein. Some controls regular MinE need to be performed, but a better choice would be to redo all the measurements since there is effectively only a single measurement. Also, mEos fluorescence overlaps with mVenus one (in a non-photoactivated state), making the choice of using this construct even more questionable.

The fusion of MinE and MinD with different fluorescent proteins has been used in many previous studies to characterize the Min oscillations, and has been proven to not affect their functionality significantly (see for example: PMID: 12766229, PMID: 11285221, PMID: 18467587). Therefore, there is no reason to suspect that our fused construct will behave differently from other fusions. On the other hand, we needed the fusion to show that there is an increase in the amount of the protein in the cell which we present in Fig. S1. As for the overlap of the fluorescence between mEos and mVenus, we agree that it does exist. However, that does not affect any of our measurements of FtsZ, because we are only measuring the accumulation of FtsZ at the ring location, which is not influenced by any possible contribution from background mEos, as it would be on average constant throughout the cell cycle.

Also, the addition of single cell measurements under different induction levels shows clearly that the cell size in that range of induction is linearly proportional to the change in the MinE/MinD ratio, which is in agreement with the model predictions (Fig. 6a). We hope that this removes any doubt regarding the functionality of the fused construct.

5) It is unclear why 65% of FtsZ intensity at the Z-ring is considered the critical threshold for cell division in *E. coli*. How would the results (Tz, Sz etc.) change if some other level is used? In any case, some justification for this threshold is needed.

We thank the reviewer for this important question. We now added the following explanation in the caption of Fig. 4: "The 65% of maximal FtsZ intensity was chosen as the threshold point, for which we measure the time needed to achieve a stable ring, because under uninduced minE conditions (Fig. 3C) we observe a clear change at that point in the rate of FtsZ accumulation within the ring." We also show in a new supplementary figure (Fig. S4) that the choice of reference intensity level does not alter the behavior we observe for Tz.

6) The rationale for presenting new data (Fig. 6) at the end of the Summary and Discussion section is unclear. It would be more meaningful to present it at the end of the Results section.

We have now added a new results section for this result: "The sister cells test".

Minor points of criticism:

1) Introduction: "The variations, though, are confined to a restricted range specific to the property being quantified ..." Not clear what point the authors aim to make. I do not see any information conveyed by this sentence.

We intended to suggest that cellular properties are well regulated and are subject to control mechanism. We now added this to the text.

2) Introduction: "Further support was obtained also by demonstrating that reducing the amount of FtsZ or increasing its degradation rate delayed cell division^{30,31}."

It does not seem that references 30, 31 are the correct ones to support this statement.

Contrary to ref 17, ref 30 states that "Our results suggest that septation is not triggered by a fixed number of newly synthesized FtsZ molecules per cell." Also, "The number of FtsZ molecules per cell can be reduced threefold without affecting the division rate significantly."

Ref 31 states, "These results support a model in which the timing of FtsZ assembly is governed primarily

through cell cycle-dependent changes in FtsZ polymerization kinetics and not simply via oscillations in the intracellular concentration of FtsZ." This is also not in support of ref 17.

None of these two works discuss degradation of FtsZ.

We have made changes there and added more references. We are not sure though about the contradiction the reviewer refers to between ref. 30 and ref. 17. Fig. 3 in ref. 30 shows that cell size increases with decreasing FtsZ abundance. However, we have removed those refs and added a new one to prevent possible confusion.

Ref. 31 was indeed the wrong ref, which we have removed. We would like to thank the referee for bringing this to our attention.

3) Introduction: "It has been demonstrated in many studies that the placement of FtsZ in the cell is determined by oscillatory dynamics of membrane-associated proteins, collectively termed the Min system^{32–40}." There are additional systems also involved in this placement and these should be mentioned. Furthermore, it should be mentioned that the Min system is not essential for the midcell placement of the Z-ring, although it prevents divisions at cell poles.

We modified the text in order to better explain this point. We also added two review articles for a more comprehensive picture of how the septum is formed, but keep our discussion focused on the Min system since it is the focus of our investigation here. The role of Min system in placing the Z-ring at mid-cell through prevention of FtsZ binding near the poles is explicitly stated at the end of the second paragraph in page 3.

4) Introduction: "This in turn allows the stable binding of FtsZ to the membrane". Unclear what "stable" means here. The residency times of FtsZ molecules on the cell membrane are about 10 seconds.

We appreciate this remark. We have now rephrased that sentence to better explain what we mean, which is that the septal ring is stable and grows continuously until division, i.e., while FtsZ might bind and unbind to the membrane, the ring is there continuously and does not disintegrate until the cell divides.

5) Introduction: "These results demonstrate the crucial effect of the fluctuations in Min proteins on the FtsZ ring formation". Unclear what "crucial" means. The cells that lack the Min system still form Z-rings and divide, yet some divisions produce minicells.

We rephrased to: "These results reveal how the Min proteins can affect cell size in E. coli by regulating the initiation time of the FtsZ ring formation."

6) Results: "We argue that this effect is mediated by the need to have an appropriate cell size to allow for a regular Min proteins dynamical pattern early in the cell cycle." Measurements should show this.

Unfortunately, we cannot measure these patterns with cell size simultaneously. Cell size changes occur over extended times, while the Min oscillations are very fast ~40 seconds. Imaging these oscillations will require acquisition rate of $\lesssim 1$ image/second. Fluorescence imaging of cells at this rate does not allow for extended measurements because it will damage the cell and will lead to premature death. That's why we opted to see this effect through the FtsZ ring formation, which can be imaged at a much lower rate. However, we note that the Min oscillations have been extensively studied, both in vitro and in vivo.

7) Fig. 2B: Not clear what the error bars show.

We now explain in the caption that the error bars are the standard deviation obtained from triplicate measurements.

8) Results: "Thus, the cell size at birth is one that allows the existing concentrations of Min proteins in the cell to produce stable patterns early in the cell cycle." The authors should show these patterns.

As we explained earlier, these patterns have been extensively studied, and measuring them here is not possible, as imaging at a high rate would damage the cell and prevent extended observation of cell size changes. We now include movies of the simulations for different Min ratios (Videos SV2 and SV3).

9) Results: "therefore cell division would be delayed until the cell reaches a new length that would allow a regular stable Min oscillation with the new concentrations."

We have rephrased this sentence.

10) Results: "Once FtsZ binds stably to the membrane, the ring starts to form, and the cell will continue to grow until the FtsZ ring is completed and the cell is ready to divide." Please clarify what "stably" means. The FtsZ in the Z-ring turns over in the time scale of 10 seconds (Stricker & Erickson PNAS 2002). Please also clarify what means "the FtsZ ring is completed". In what sense the FtsZ ring is completed?

We have rephrased this sentence which now reads: "Once FtsZ accumulation at the membrane is uninterrupted, the ring becomes stable and grows continuously, and the cell will grow until the FtsZ ring can initiate constriction and cell division."

11) "Our observations show that immediately following the induction of minE-mEos expression, cells become longer". Unclear what is meant. I assume cells continue to grow.

We changed it to: "Our observations show that immediately following the induction of minE-mEos expression, cells grow to a longer size than in previous cell cycles before dividing..."

12) Fig. 3B : "Final cell size" - I would call it cell length at division. "Elongating" – after induction (are there cells that are not elongating?). What do the error bars show?

We have changed "Final cell size" to "Cell length at division", and "elongating" to "induced". The error bars represent the standard error in the measurement. This has now been added to the figure caption.

13) Fig. 3C-D: FtsZ intensity at midcell should decrease at the end of the cell cycle, but this decrease is missing, indicating some problems in the analysis.

Yes. For slow growth in limited medium (like here doi: 10.1016/j.cub.2019.12.02) where cell cycle time ~90 min (3x our case) the decay happens in the last 5 mins of the cycle (last ~ 5% of the total cell cycle). In our case the last 5% is ~1.5 min which is close to the frame acquisition rate (1 min). This bars us from capturing the decay of ftsZ in all division events. That, however, does not affect our measurements since we are only interested in identifying the formation of stable rings.

14) ftsZ -> FtsZ. What does "stable" mean?

Explained above.

15) Fig. 4B-G: Why are the error bars based on bootstrapping? The reader deserves to see std or std errors based on conventional calculation.

The standard deviation represents the variation in cell size, which is very large relative to the change in the average cell size. The error bars now represent the standard error. However, we also added the distribution of sizes for different time points in Fig. S4.

16) Fig. 4: Referring "normal" before induction can be misleading. Potentially [MinE]/[MinD] is significantly higher than in WT cells. The authors should mention what is [MinE]/[MinD] before induction. The applicability of determining this ratio using RT-PCR should also be discussed.

Changed to uninduced.

17) Summary and Discussions: "These experiments revealed that disturbing the balance of the Min proteins delays FtsZ binding to the membrane and the Z-ring formation initiation until the cell reaches a size that allows for a stable regular dynamical pattern of the Min proteins with the new concentrations" I cannot find these experiments described in the manuscript.

We have rephrased this sentence to separate between conclusions and experimental results.

18) Summary and Discussions: "More specifically, since the concentrations of Min proteins that sister cells receive from their mother are similar, their sizes at the onset of the FtsZ ring formation should be similar." Not clear if this assumption is correct, taking that the Min system shows no "stable regular dynamic" pattern at early stages of the cell cycle. Furthermore, the assumptions the authors make about FtsZ distribution between daughter cells need to be explained.

We have rephrased that sentence, and added a reference that shows that the distribution of Min proteins between sister cells is close to symmetric. As for the distribution of the FtsZ proteins, we do not make any assumption. We simply assume that the initiation of the FtsZ ring formation is not limited by the amount of FtsZ in the cell at birth. Of course, this assumption is for the average behavior of the population, which is supported by our observations.

19) The authors' model is effectively a sizer-model with a critical cell cycle checkpoint when the FtsZ at the Z-ring reaches 65% of its maximum level. Interesting, but the evidence comes from a single poorly defined measurement (MinE-mEos upregulation). There is a large body of literature based on much more explicit measurements that refute sizer models.

As we stated previously, our model is not a sizer model. In fact, we argue that it is an adder model as previously shown if the measurement of the added size is taken from the onset of the FtsZ ring formation (Fig. 4E), in agreement with previous measurements in which the accumulation of FtsZ (and not DNA replication) is the bottleneck process for cell division. Note that the adder phenomenon is seen only on average at the population level, while individual cells still exhibit large variations and deviations from

that behavior. What we propose here is that part of this deviation is the result of the additional check point by the Min proteins early in the cell cycle. This check point under constant conditions and assuming the cell division is exactly symmetric, will not change the adder phenomenon, as it will only add a very small constant size to the total added size, which is acquired within the first few minutes of the cell cycle. If a cell divides asymmetrically, however, then one cell will add a slightly larger size than the other. This again will not change the adder phenomenon because when averaging over the population, the sisters will cancel each other, and each cell would seem as if it added a constant fixed size on average (see Fig. S7).

We also included new measurements under different growth conditions and induction levels of minE, which provide further support to our arguments.

20) Fig. 6B: Based on large error bars, it is unclear if there is a significant trend. The authors present no statistical test. Also, there is no explanation of what the error bars show, how many cells were analyzed, what conditions the experiments were done, etc.

We now add in the caption the number of cells used, and how the error bars were calculated. We also added a new supplementary figure (Fig. S7), that uses larger number of measurements from previous papers on sister cells. The new figure clearly demonstrates the trend in added length of sister cells, and how such data still exhibits an adder phenomenon at the population level.

Reviewer #1 (Remarks to the Author):

I am generally happy with the revisions of this manuscript, except that it seems to me that one of the points that I previously raised was not really addressed, or not completely.

Specifically, when asking about single cell behavior, I might not have been sufficiently clear in saying that I would like to see to single-cell cycle-size correlation plots (such as adder plots, there are many variants and different correlations to look and the authors have themselves addressed this issue in PMID: 33357764, where the data in Fig. S7 comes from)

- before the shift (this should be equivalent to existing literature so not very exciting)
- in steady-dividing cells with MinE overexpression
- Along the shifts in Fig 4 and S3-S5, see for example Fig 2E of Panlilio et al, ref. 29.

The authors argue that the adder correlation pattern is always stable, and I am inclined to believe that this should be the case, but as far as I could see they did not really show it with their data. As far as I understood the (unperturbed) data in Fig S7 come from a different study - which appears a bit strange. If there are problems with such plots with data from this study, then the issue should emerge - and it should be discussed. Otherwise it would seem very natural to see these plots.

Fig 5B is also interesting (although it is a property of the simple accumulator models also proposed and used in previous studies), but can the authors compare its behavior before vs after minE overexpression, as well as during the transient? [As a minor point it would be nice to use compatible notations in for Fig. 5 and S7.]

Obviously it would also be nice to correlate this behavior with FtsZ / minE in single cells (expanding on Fig S3), but I am not asking this.

I also have a possible suggestion for the discussion. The authors more or less that E. coli cells have a way of modulating their population-average size, at fixed population-average growth rate, as well as at fixed size-control strategy (and scaling propoertied of the distribution). This can be used to formulate hypotheses on the evolutionary plasticity and role of cell size. There is a fairly large (and not very organic) body of literature on this, ranging from hypotheses from the cell-size control literature, to findings in laboratory evolution experiments, to studies and hypotheses coming from the ecology and quantitative ecology angle. Perhaps this could expand the scope of the last paragraph of the discussion.

Reviewer #2 (Remarks to the Author):

The revised manuscript by Vashistha et al. has addressed some points of my criticism but not adequately the two main comments. A description of the model for the Min oscillations remains also missing.

Main points of criticism:

1) The manuscript still lacks experimental data on how the distribution of MinD changes at different concentrations of MinE and at different cell lengths. Without these measurements, the main conclusions are not backed up by experimental data and remain highly speculative.

2) It is still not convincing that the 3-fold upregulation of [MinE]/[MinD] can be extrapolated to physiological conditions. The data in Fig. 6a is sketchy. The R/R0 ratios are all very close together and the data does not appear linear, contrary to the claims by the authors. The statistics for these data points are also rather poor (20-40 cells).

3) There is no description of the model for the Min oscillations. A scarcely commented code is no substitute for the description of the model.

4) There is no description of how the parameters of the Min oscillations are chosen and/or adjusted for the curves in Fig. 6.

5) Fig. 6 caption, "Note that while the agreement between the dynamics observed in simulations and experiments is quantitative, only qualitative agreement is observed between them in the actual time and size measurements. This is due to the fact that the cell size that allows for stable FtsZ ring formation in the simulations depends on the kinetic parameters used." This statement is unclear and needs to be explained.

6) The term "stable Z-ring formation" needs to be better justified. It seems to be 65% FtsZ present in the Z ring of its maximum value that is used to define stable Z ring formation. It is unclear how this operational definition relates to the stability of the Z ring.

Minor points:

1) In several passages, including the abstract, the authors mention:

"the cell needs to reach a size that can accommodate a new dynamical pattern of the Min proteins (at the new concentrations) for which MinCD occupancy at mid-cell is low enough to allow a stable FtsZ ring formation."

What is this new dynamical pattern? Assuming the model is correct, there is no change in the pattern (Fig. S8).

2) In Fig. S8D, "The average time fraction during which the mid-cell is free of MinD as a function of cell size." This statement cannot be correct. There is some MinD concentration at midcell at all times. What threshold was chosen and why? How the outcome depends on this threshold?

3) Fig. 6 caption, "Error bars depict the sampling resolution we used to determine the optimal cell size in the simulations." Unclear what the sampling resolution means here.

4) "The sister cell test". It comes in between two conceptually connected sections now. It could fit better as the last section of the Results.

Dear Editor,

We would like to thank the reviewers again for their feedback, which we address in detail below. All the changes we made to the main text, supporting material, including figures' numbers due to rearrangements we had to make to satisfy the reviewers requests, are labeled in red for easier tracking. We hope that the reviewers will now approve the manuscript for publication.

On behalf of all authors,

Hanna Salman

Reviewer #1 (Remarks to the Author):

I am generally happy with the revisions of this manuscript, except that it seems to me that one of the points that I previously raised was not really addressed, or not completely.

Specifically, when asking about single cell behavior, I might not have been sufficiently clear in saying that I would like to see to single-cell cycle-size correlation plots (such as adder plots, there are many variants and different correlations to look and the authors have themselves addressed this issue in PMID: 33357764, where the data in Fig. S7 comes from)

- before the shift (this should be equivalent to existing literature so not very exciting)
- in steady-dividing cells with MinE overexpression
- Along the shifts in Fig 4 and S3-S5, see for example Fig 2E of Panlilio et al, ref. 29.

The authors argue that the adder correlation pattern is always stable, and I am inclined to believe that this should be the case, but as far as I could see they did not really show it with their data. As far as I understood the (unperturbed) data in Fig S7 come from a different study - which appears a bit strange. If there are problems with such plots with data from this study, then the issue should emerge - and it should be discussed. Otherwise it would seem very natural to see these plots.

We thank the reviewer for approving our revision. We agree that we misunderstood the reviewer previous point regarding the adder test. We have now provided these tests. In the new Figure S9, which we address also in the main text lines 284 – 286, we show the added size as a function of the birth size before, during and post induction. Our results show that the added size is, to good extent, independent of the birth size, throughout the induction process.

Fig 5B is also interesting (although it is a property of the simple accumulator models also proposed and used in previous studies), but can the authors compare its behavior before vs after minE overexpression, as well as during the transient? [As a minor point it would be nice to use compatible notations in for Fig. 5 and S7.]

We attach here this comparison for the reviewer evaluation.

However, we don't really see any value in adding this comparison to the paper. Post induction the behavior should be similar to prior induction because the cell is in steady state so it does not provide any additional information. And during induction the cell length at division is continuously changing, and we have no theory to how the sisters should behave, and we fear that adding these graphs might confuse the reader.

Obviously it would also be nice to correlate this behavior with FtsZ / minE in single cells (expanding on Fig S3), but I am not asking this.

We are not sure what correlation is the reviewer asking about, but we appreciate that he/she is not requesting it for this paper. If the reviewer's intention is the correlation between FtsZ / MinE distribution between sisters and the added size of sisters, we agree it will be interesting to study, and as we mention at the end of the discussion, this will require further investigation.

I also have a possible suggestion for the discussion. The authors more or less that E. coli cells have a way of modulating their population-average size, at fixed population-average growth rate, as well as at fixed size-control strategy (and scaling properties of the distribution). This can be used to formulate hypotheses on the evolutionary plasticity and role of cell size. There is a fairly large (and not very organic) body of literature on this, ranging from hypotheses from the cell-size control literature, to findings in laboratory evolution experiments, to studies and hypotheses coming from the ecology and quantitative ecology angle. Perhaps this could expand the scope of the last paragraph of the discussion.

We now address this in the discussion, lines 420 – 426, as requested.

Reviewer #2 (Remarks to the Author):

The revised manuscript by Vashistha et al. has addressed some points of my criticism but not adequately the two main comments. A description of the model for the Min oscillations remains also missing.

Main points of criticism:

1) The manuscript still lacks experimental data on how the distribution of MinD changes at different concentrations of MinE and at different cell lengths. Without these measurements, the main conclusions are not backed up by experimental data and remain highly speculative.

We are not sure which conclusions the reviewer means. The main conclusion of this study is that changing the expression level of the Min proteins changes the cell size. We have demonstrated this by directly changing the expression of MinE and measuring the resulting ratio of MinE/MinD and the effect on the cell size, both at the single-cell and population levels, and under different experimental conditions. Nowhere in the manuscript do we make any claim about the distribution of MinD and its effects on cell size under different MinE expression levels and in different cell lengths. If the reviewer point is about our suggestion that variability in cell size could result from fluctuations in MinE and MinD concentrations, then we need to stress that this is indeed a speculation based on the results of this study.

Since we have seen that changing the expression ratio of MinE/MinD can affect cell size, it is reasonable to suggest that the variation in cell size measured under normal conditions could in part be due to fluctuations in the distribution of MinE and MinD between cells and fluctuations in their expression over time. Again, we stress that we are merely suggesting this and clearly stating that it requires further testing at the end of the discussion section: "Further experiments are needed to address the contribution of the Min proteins to the variability in size homeostasis within a population, in which the ratio of the Min proteins could be measured at the single-cell level and in different conditions, and their correlation with cell size could be evaluated." We also changed the end of the abstract to make it clear that our results may explain this variability, but this remains to be tested.

Therefore, we think that this request is not relevant to the current study, and these measurements constitute an independent study for the future.

2) It is still not convincing that the 3-fold upregulation of [MinE]/[MinD] can be extrapolated to physiological conditions. The data in Fig. 6a is sketchy. The R/R0 ratios are all very close together and the data does not appear linear, contrary to the claims by the authors. The statistics for these data points are also rather poor (20-40 cells).

First, we would like to stress that our choice of the number of experiments was based on the fact that we did not find any statistically significant change in the results when we added cells past 20. Nevertheless, we have now included further data to support our claims. We have measured the change in cells size at the population level for several induction levels, for which the ratio MinE/MinD was also measured, and repeated the measurement several times. These new results are included in the new Fig. 2 and Fig. S7. It can be clearly seen that both single-cell and population measurements are in agreement with each other.

Nevertheless, we agree with the reviewer that the change in size is not necessarily linear with the change in the ratio MinE/MinD, and we make it clear now in the manuscript (lines 314 – 316, and Fig. 5 caption line 324) that this is a first-order approximation. However, in a first-order approximation a linear fit is good enough, and more importantly, it is sufficient to describe the dynamics of ring formation and size change following the induction of MinE overexpression. This is the main point that we were trying to make here, that a linear description of the change in size as a function of the change in the ratio MinE/MinD (which is normally the first to be tested) is sufficient to reproduce the measured dynamics following MinE overexpression induction. Since first-order approximation is sufficient to describe the measured dynamics, there is no need to use more elaborate or non-linear description of the data. And again, the extrapolation of the data to fluctuations in the physiological conditions is a speculation that we make based on our results. As we mentioned earlier, we speculate that these results could explain part of the variability in cell size measured under homogenous conditions, and we make it clear that this is a speculation that requires further testing.

3) There is no description of the model for the Min oscillations. A scarcely commented code is no substitute for the description of the model.

We have now added a description of the model in the supplementary Information before the simulation code (lines 35 – 61).

4) There is no description of how the parameters of the Min oscillations are chosen and/or adjusted for the curves in Fig. 6.

The parameters used and their source are included now in the new model description in the Supplementary Information. The only change that was done in the simulations used in Fig. 6 (now Fig. 5) was the concentration of MinE, which would result in a change in the ratio MinE/MinD as depicted in the graphs. For each of the MinE values, the optimal cell length was obtained as described in Fig. S10 and presented in Fig. 5A. For the rest of the graphs in Fig. 5, we have used exponential bacterial growth dynamics assuming a constant elongation rate. We also used gfp expression to model the increase of minE protein over time and used that to calculate the change in the ratio (R/R0) as function of time presented in Fig. 5B. This is described in lines 312 – 346. We then used Figs. 5A and B to calculate Tz and S0 presented in Figs. 5C and D respectively, as explained in lines 347 – 364.

5) Fig. 6 caption, “Note that while the agreement between the dynamics observed in simulations and experiments is quantitative, only qualitative agreement is observed between them in the actual time and size measurements. This is due to the fact that the cell size that allows for stable FtsZ ring formation in the simulations depends on the kinetic parameters used.” This statement is unclear and needs to be explained.

The simulations we carried out were based on previous studies in which the parameters were chosen to create stable oscillations with minimal occupancy at the mid-cell for a cell of length 4.5 μm , which is obviously not the size of the cells in the experiments. Therefore, the comparison can be only qualitative, since we do not have a direct measurement of the kinetic parameters of the different reactions involved in the Min interactions. We state this now in the caption, lines 335 – 336.

6) The term “stable Z-ring formation” needs to be better justified. It seems to be 65% FtsZ present in the Z ring of its maximum value that is used to define stable Z ring formation. It is unclear how this operational definition relates to the stability of the Z ring.

In the first paragraph of the results section Effect of MinE overexpression on the binding time of FtsZ, we define a stable Z-ring (lines 196 – 198): “Once FtsZ accumulation at the membrane is uninterrupted, the ring becomes stable and grows continuously...”. The reason for the choice of 65% is due to the fact that we do not see FtsZ ring falling apart for intensities higher than ~50% of max, which reflects that the ring is stable. Figure 4A shows transient ring formation for which the intensity is much lower than the threshold we use here. We further justify our use of the 65% threshold in the caption of figure 4 but also show in figure S5, that the choice of the cutoff does not affect the results.

Minor points:

1) In several passages, including the abstract, the authors mention:

“the cell needs to reach a size that can accommodate a new dynamical pattern of the Min proteins (at the new concentrations) for which MinCD occupancy at mid-cell is low enough to allow a stable FtsZ ring formation.”

What is this new dynamical pattern? Assuming the model is correct, there is no change in the pattern (Fig. S8).

Change in dynamical pattern does not refer to the shape of the wave only, but any change in how the wave sweeps the cell length. That includes wavelength relative to cell length, wave speed etc. In figure S8 we show the probability of occupancy along the cell length, which is clearly different between the different conditions. This is a result of how the MinCD wave sweeps the cell and at what frequency. This is also explained in the last paragraph of the results section “Increased expression of MinE increases cell size” (lines 182 – 183) and in the second paragraph of the results section “Experimental results confirm model predictions”, Lines 304 – 309.

2) In Fig. S8D, “The average time fraction during which the mid-cell is free of MinD as a function of cell size.” This statement cannot be correct. There is some MinD concentration at midcell at all times. What threshold was chosen and why? How the outcome depends on this threshold?

We now provide a better explanation of how that graph was obtained in the caption of the figure.

3) Fig. 6 caption, “Error bars depict the sampling resolution we used to determine the optimal cell size in the simulations.” Unclear what the sampling resolution means here.

The simulations for each MinE/MinD ratio were carried out at different cell sizes. The difference between the sizes used is the sampling resolution, which was considered as the error in determining the optimal size.

4) “The sister cell test”. It comes in between two conceptually connected sections now. It could fit better as the last section of the Results.

We moved that section to the end of the results.

Reviewer #1 (Remarks to the Author):

I looked at the manuscript and I am happy with the revision concerning my comments.

Reviewer #2 (Remarks to the Author):

In response to my main comment on the previous version of the manuscript, the authors write: "We are not sure which conclusions the reviewer means. The main conclusion of this study is that changing the expression level of the Min proteins changes the cell size. We have demonstrated this by directly changing the expression of MinE and measuring the resulting ratio of MinE/MinD and the effect on the cell size, both at the single-cell and population levels, and under different experimental conditions."

"The main conclusion of this study is that changing the expression level of the Min proteins changes the cell size." has been known for over 30 years. See, for example, de Boer et al. Cell 1989.

I took the main conclusion to be, "Our analyses at the single-cell level reveal that the altered ratio of Min proteins, specifically increasing MinE/MinD, delays the FtsZ ring formation until the cell reaches a size that facilitates a regular dynamical pattern of the Min proteins, which permits a continuous accumulation of FtsZ at the membrane and the subsequent formation of a stable septal ring that does not disintegrate until division." (Lines 108-113).

Again, there was no backing in the previous nor the current version of the manuscript that "...the cell reaches a size that facilitates a regular dynamical pattern of the Min proteins, which permits a continuous accumulation of FtsZ at the membrane". From the authors' responses, I gather that the dynamical pattern does not change in their model. Even if it changed, there is no indication that the model is correct. There is no experimental data on coefficients entering the model. Even if the reactions in the model are correctly described, the reaction rates can be orders of magnitude off. Even units for these coefficients are not correct in SI Text. The unit for the diffusion coefficient is micrometer²/s, not micrometer/s. There is also a discrepancy between sigma(dD) and sigma(E) units.

Further on conclusions:

Line 428 "Importantly, they reveal that cell size is sensitive to internal cellular gene expression ratios."

I do not see this work revealing "...that cell size is sensitive to internal cellular gene expression ratios". The MinE/MinD ratio in the wild-type cells does not vary in the range probed in these experiments. MinE and MinD are expressed from the same operon, and their ratio is well controlled. For other gene ratios, it remains to be seen how sensitive it is for the division process. Many division-related genes are expressed from a single operon, and their ratios are well controlled.

REVIEWERS' COMMENTS

Reviewer #1 (Remarks to the Author):

I looked at the manuscript and I am happy with the revision concerning my comments.

We thank the reviewer for supporting the publication of our manuscript.

Reviewer #2 (Remarks to the Author):

In response to my main comment on the previous version of the manuscript, the authors write: “We are not sure which conclusions the reviewer means. The main conclusion of this study is that changing the expression level of the Min proteins changes the cell size. We have demonstrated this by directly changing the expression of MinE and measuring the resulting ratio of MinE/MinD and the effect on the cell size, both at the single-cell and population levels, and under different experimental conditions.”

“The main conclusion of this study is that changing the expression level of the Min proteins changes the cell size.” has been known for over 30 years. See, for example, de Boer et al. Cell 1989.

I took the main conclusion to be, “Our analyses at the single-cell level reveal that the altered ratio of Min proteins, specifically increasing MinE/MinD, delays the FtsZ ring formation until the cell reaches a size that facilitates a regular dynamical pattern of the Min proteins, which permits a continuous accumulation of FtsZ at the membrane and the subsequent formation of a stable septal ring that does not disintegrate until division.” (Lines 108-113).

We agree with the reviewer assessment.

Again, there was no backing in the previous nor the current version of the manuscript that “...the cell reaches a size that facilitates a regular dynamical pattern of the Min proteins, which permits a continuous accumulation of FtsZ at the membrane”. From the authors’ responses, I gather that the dynamical pattern does not change in their model. Even if it changed, there is no indication that the model is correct. There is no experimental data on coefficients entering the model. Even if the reactions in the model are correctly described, the reaction rates can be orders of magnitude off. Even units for these coefficients are not correct in SI Text. The unit for the diffusion coefficient is micrometer²/s, not micrometer/s. There is also a discrepancy between $\sigma(dD)$ and $\sigma(E)$ units.

The support we have for our assumption that the cell reaches a size that allows for the FtsZ ring formation to proceed uninterrupted is that when we disturb the MinE/MinD ratio the FtsZ ring does not form continuously until the cell reaches a larger size. Further, by comparing our measured dynamics of size increase and FtsZ ring formation time with the predictions we obtain from the simulations we find a good agreement between the two. As for the change in the dynamical pattern, as we explain in the text lines 164 – 169 and 247 – 250, the change is in the sweeping pattern and frequency of the membrane by the MinCD, which can interfere in the accumulation of the FtsZ ring. Indeed, we do not claim that the

shape of the sweeping wave changes, however, the change in the frequency and the occupancy of the membrane by MinCD affects the average distribution of MinCD along the cell and that does affect the buildup of the FtsZ ring.

We have now clarified this further in the introduction (lines 110 – 117) and discussion (lines 353 – 367) sections.

Further on conclusions:

Line 428 “Importantly, they reveal that cell size is sensitive to internal cellular gene expression ratios.”

I do not see this work revealing “...that cell size is sensitive to internal cellular gene expression ratios”. The MinE/MinD ratio in the wild-type cells does not vary in the range probed in these experiments. MinE and MinD are expressed from the same operon, and their ratio is well controlled. For other gene ratios, it remains to be seen how sensitive it is for the division process.

Many division-related genes are expressed from a single operon, and their ratios are well controlled.

What we meant by that sentence is that there are multiple factors that can influence cell size. We have rephrased this now and made it more specific to MinE/MinD ratio. And we further clarified that this is based on tests done beyond the physiological range of MinE/MinD ratio, and the effect in the physiological range is speculative that need to be experimentally tested in the future.